# *ACVR1* R206H cooperates with H3.1K27M in promoting diffuse intrinsic pontine glioma pathogenesis

Christine M. Hoeman[1], Francisco J. Cordero[2], Guo Hu[3], Katie Misuraca[4], Megan M. Romero[1], Herminio J. Cardona[1], Javad Nazarian[5], Rintaro Hashizume[6,7,8], Roger McLendon[9,10], Paul Yu[11], Daniele Procissi[12], Samantha Gadd[13] & Oren J. Becher[1,7,8,14]

Diffuse intrinsic pontine glioma (DIPG) is an incurable pediatric brain tumor, with approximately 25% of DIPGs harboring activating *ACVR1* mutations that commonly co-associate with H3.1K27M mutations. Here we show that in vitro expression of *ACVR1* R206H with and without H3.1K27M upregulates mesenchymal markers and activates Stat3 signaling. In vivo expression of *ACVR1* R206H or G328V with H3.1K27M and p53 deletion induces glioma-like lesions but is not sufficient for full gliomagenesis. However, in combination with PDGFA signaling, *ACVR1* R206H and H3.1K27M significantly decrease survival and increase tumor incidence. Treatment of *ACVR1* R206H mutant DIPGs with exogenous Noggin or the *ACVR1* inhibitor LDN212854 significantly prolongs survival, with human *ACVR1* mutant DIPG cell lines also being sensitive to LDN212854 treatment. Together, our results demonstrate that *ACVR1* R206H and H3.1K27M promote tumor initiation, accelerate gliomagenesis, promote a mesenchymal profile partly due to Stat3 activation, and identify LDN212854 as a promising compound to treat DIPG.

[1] Department of Pediatrics, Northwestern University, Chicago, IL 60611, USA. [2] GI Oncology Research Unit, Duke Cancer Institute, Duke University, Durham, NC 27710, USA. [3] Department of Molecular and Human Genetics, Baylor College of Medicine, Houston, TX 77030, USA. [4] Department of Pediatrics, Duke University, Durham, NC 27710, USA. [5] Department of Integrative Systems Biology, Children's National Medical Center, George Washington University, Washington, DC 20010, USA. [6] Department of Neurosurgery, Northwestern University, Chicago, IL 60611, USA. [7] Department of Biochemistry and Molecular Genetics, Northwestern University, Chicago, IL 60611, USA. [8] Lurie Comprehensive Cancer Center, Northwestern University, Chicago, IL 60611, USA. [9] Department of Pathology, Duke University Medical Center, Durham, NC 27710, USA. [10] Preston Robert Tisch Brain Tumor Center, Duke University Medical Center, Durham, NC 27710, USA. [11] Department of Medicine, Cardiovascular Division, Brigham and Women's Hospital, 75 Francis Street, Boston, MA 02115, USA. [12] Department of Radiology, Northwestern University, Chicago, IL 60611, USA. [13] Department of Pathology, Ann & Robert H. Lurie Children's Hospital, Chicago, IL 60611, USA. [14] Division of Hematology Oncology and Stem Cell Transplant, Ann & Robert H. Lurie Children's Hospital, Chicago, IL 60611, USA. Correspondence and requests for materials should be addressed to O.J.B. (email: oren.becher@northwestern.edu)

Diffuse intrinsic pontine glioma (DIPG) originates in the pons and comprises approximately 20% of all pediatric brain tumors[1,2]. DIPG remains incurable with a median survival of 10–11 months[3]. Failure to identify a successful therapy for DIPG likely stems from the lack of biological understanding of the disease, as until recently biopsies were not commonly performed due to the sensitive location of the tumor. As a result, many trials have been based upon genetic alterations found in adult glioblastomas. Recent studies have shown that DIPG is molecularly distinct from adult gliomas. High-throughput sequencing unraveled mutations in genes encoding histone variants H3.3 and H3.1 (*H3F3A*, *HIST1H3B*, and *HIST1H3C*) in ~80% of all DIPG patients[4–7], resulting in the substitution of a lysine to a methionine at position 27 (K27M), and leading to abnormal Polycomb Repressive Complex 2 (PRC2) function. As a result, H3K27 methylation is reduced globally but focally gained[8–11]. H3K27M mutations are found primarily in midline high-grade gliomas and rarely occur in pediatric high-grade gliomas located in the cerebral cortex or in adult high-grade gliomas[12].

Since the finding of H3.3 and H3.1K27M mutations in DIPG, we and others have discovered that ~25% of DIPG patients harbor activating *ACVR1* mutations, a gene that encodes for the ALK2, a receptor in the bone morphogenetic protein (BMP) signaling pathway[6,13–15]. *ACVR1* mutations were found to activate the BMP signaling pathway independent of receptor ligation and commonly segregate with H3.1K27M mutations. Surprisingly, *ACVR1* mutations found in DIPG were nearly identical to those found in the connective tissue disease fibrodysplasia ossificans progressiva (FOP), an autosomal dominant disease in which progenitor populations within muscles, tendons, and ligaments undergo heterotopic ossification or ectopic bone formation[16,17]. Interestingly, FOP patients do not develop DIPG, suggesting that *ACVR1* mutations alone are not sufficient to drive gliomagenesis.

However, the mechanism underlying the effect of *ACVR1* and H3.1K27M mutations on gliomagenesis in the context of DIPG remains to be determined. To date no one has shown whether *ACVR1* mutations, either individually or in combination with H3.1K27M mutations, are oncogenic drivers and can lead to gliomagenesis. In this study, we use the RCAS/tv-a system to study the effects of these mutations in vitro and in vivo. We observe that *ACVR1* mutations have differential effects on proliferation and survival in vitro with *ACVR1* R206H being the most potent. RNAseq analysis of nestin-expressing brainstem progenitors infected with *ACVR1* R206H demonstrates that *ACVR1* R206H upregulates the expression of mesenchymal markers and activates Stat3 signaling relative to *ACVR1* WT with and without H3.1K27M. While *ACVR1* mutations can lead to the formation of glioma-like lesions, with H3.1K27M and p53 loss being required for this process, such genetic alterations are not sufficient to drive tumor development. Rather, PDGFA signaling is required for full gliomagenesis, and in the presence of PDGFA signaling and p53 loss, both *ACVR1* R206H and H3.1K27M accelerate gliomagenesis. Additionally, the combination of *ACVR1* R206H and H3.1K27M increases tumor incidence, suggesting that these mutations play a role in tumor initiation. Lastly, we identify the BMP signaling pathway as a potential effective therapeutic strategy to treat *ACVR1* R206H mutant DIPGs as exogenous Noggin expression at tumor initiation or treatment with LDN212854 significantly increases tumor latency and prolongs survival. We validate our observations in human models by demonstrating that human *ACVR1* mutant DIPG cell-lines are sensitive to treatment with LDN212854 in vitro and that human tumors with *ACVR1* mutations harbor increased Stat3 signaling. In conclusion, this study highlights the role of mutant *ACVR1*

and H3.1K27M in tumor initiation and identifies the BMP pathway, particularly LDN212854, as a promising therapeutic agent for further evaluation in *ACVR1* mutant DIPG.

## Results

**ACVR1 mutations affect proliferation and survival.** Previously we have examined *ACVR1* mutations in vitro using the RCAS/tv-a system with brainstem progenitor cells cultured as adherent cell lines[6]. To study the effects of *ACVR1* mutations on stem-like progenitors, cells were isolated from the brainstem of postnatal day 3 (p3) Nestin-Tv-a (Ntv-a);p53$^{fl/fl}$ mice, cultured as neurospheres, and infected with RCAS-*ACVR1* WT, R206H, G328V, or G328E virus. Infection with *ACVR1* R206H virus significantly increased proliferation relative to *ACVR1* G328V and *ACVR1* G328E (Fig. 1a) and cell survival relative to *ACVR1* WT and *ACVR1* G328E (Fig. 1b). To determine BMP signaling pathway activation, protein levels of phosphorylated SMAD1/5/8 and Id1, a downstream target of BMP signaling, were assessed. Infection with all *ACVR1* mutations led to increased phosphorylated SMAD1/5/8 expression as compared to *ACVR1* WT, with *ACVR1* G328E increasing pSMAD1/5/8 modestly but significantly relative to *ACVR1* WT (Fig. 1c). Id1 levels were also significantly upregulated by infection with *ACVR1* mutations relative to *ACVR1* WT with *ACVR1* G328V increasing Id1 protein levels the most (Fig. 1d). In summary, all three *ACVR1* mutations increase the activation of the BMP pathway with *ACVR1* R206H having the most effect on proliferation and survival.

Since *ACVR1* mutations commonly co-segregate with H3.1K27M mutations, we hypothesized that the two mutations would have an additive effect on these results. Surprisingly, neither proliferation nor cell survival levels were significantly increased when neurospheres were infected with *ACVR1* and H3.1K27M mutations as compared to neurospheres infected with *ACVR1* mutations alone (Supplementary Figure 1). SMAD1/5/8 phosphorylation and Id1 protein levels increased slightly with the addition of H3.1K27M but were not significant (Supplementary Figure 1). To determine how *ACVR1* mutations were behaving individually and in combination with H3.1K27M mutations, RNASeq analysis was performed on neurospheres that were infected with *ACVR1* R206H or *ACVR1* WT, both in the presence and absence of H3.1K27M (Fig. 1e). Surprisingly, only 24 genes were significantly differentially expressed by *ACVR1* R206H relative to *ACVR1* WT without H3.1K27M while 2478 genes were significantly differentially expressed by *ACVR1* R206H relative to *ACVR1* WT in the presence of H3.1K27M (*p* adjusted < 0.05). As an example, all 4 ID genes were significantly upregulated by *ACVR1* R206H in the presence of H3.1K27M while only ID1 was significantly upregulated by *ACVR1* R206H in the absence of H3.1K27M (Supplementary Data 1). Furthermore, GSEA analysis showed a significant enrichment of genes involved in epithelial to mesenchymal transition (EMT) as well as the IL-6/JAK/STAT3 signaling pathway both with and without H3.1K27M (Fig. 1f, g, Supplementary Figure 1, and Supplementary Data 2), suggesting that *ACVR1* R206H is primarily promoting these signatures and that H3.1K27M enhances them. These results were confirmed with *ACVR1* R206H having increased expression of Id1 and Socs3, a STAT3 target gene and one of the 24 significantly differentially upregulated genes, as compared to *ACVR1* WT alone (Supplementary Figure 1). Furthermore, increased mesenchymal gene expression of CD44, tenascin C (TNC), and Snail2 and downregulation of proneural genes such as Sox10, was observed, suggesting that *ACVR1* mutations might contribute to a mesenchymal phenotype in the presence of the mutant histone (Fig. 1h). In light of these results and as STAT3 signaling has been shown to regulate mesenchymal genes in brain tumors[18], we

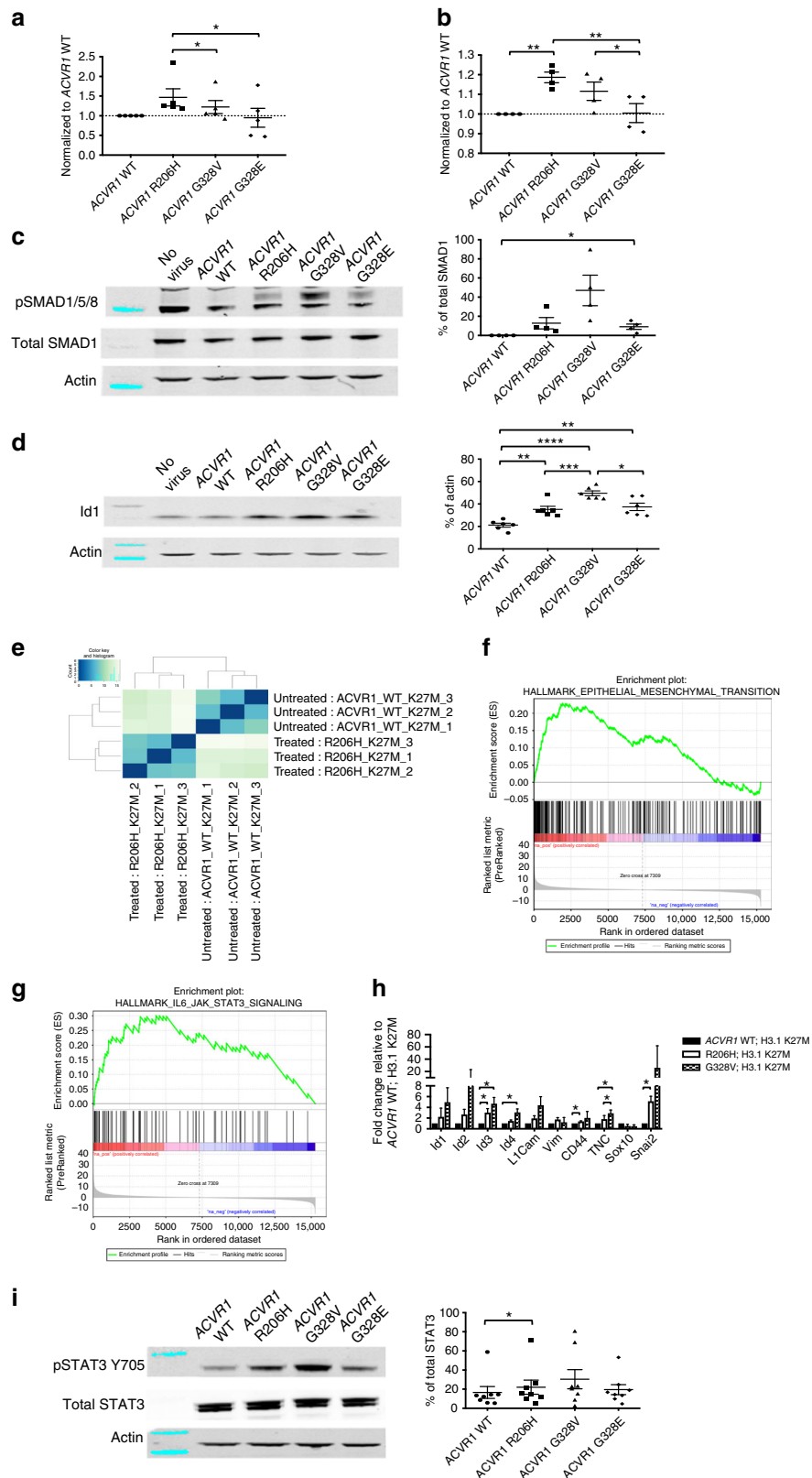

examined whether STAT3 signaling was activated by *ACVR1* mutations at the protein level. Indeed, cells that were infected with *ACVR1* mutations had modestly increased levels of phosphorylated STAT3 Y705 as compared to *ACVR1* WT infected cells, with *ACVR1* R206H demonstrating significantly increased levels (Fig. 1i). However, infection with *ACVR1*

mutations and H3.1K27M did not significantly increase phosphorylated STAT3 Y705 expression compared to infection with *ACVR1* mutations alone (Supplementary Figure 1). Together, these results indicate that one way *ACVR1* mutations contribute to DIPG pathogenesis is by promoting a mesenchymal profile which may be due in part to increased STAT3 signaling while

**Fig. 1** *ACVR1* mutations have differential effects on proliferation and survival. **a–d** Brainstem progenitor cells isolated from p3 Ntv-a;p53$^{fl/fl}$ mice were cultured in vitro as neurospheres and infected with *ACVR1* WT, *ACVR1* R206H, *ACVR1* G328V, or *ACVR1* G328E virus. **a** Proliferation of infected neurospheres ($n = 5$). **b** Cell viability of infected neurospheres ($n = 4$). **c, d** pSMAD1/5/8 (**c**) ($n = 4$) and Id1 (**d**) ($n = 6$) protein expression in infected neurospheres. Also see Supplementary Figure 1. **e–h** Brainstem progenitor cells isolated from p3 Ntv-a;p53$^{fl/fl}$ mice were cultured in vitro as neurospheres and infected with *ACVR1* WT or R206H virus in the presence of H3.1K27M virus. **e** Heat map of differentially expressed genes between *ACVR1* WT and H3.1K27M infected neurospheres compared to *ACVR1* R206H and H3.1K27M infected neurospheres based on $p$ value identified by RNA-Seq analysis. Also see Supplementary Data 1. **f** GSEA enrichment plot of epithelial to mesenchymal transition genes identified by RNA-Seq analysis from (**e**). Also see Supplementary Data 2. **g** GSEA enrichment plot of IL-6_JAK_STAT3 signaling genes identified by RNA-Seq analysis from (**e**). Also see Supplementary Data 2. **h** qRT-PCR validation of select genes from neurospheres infected with *ACVR1* WT, *ACVR1* R206H, or *ACVR1* G328V and H3.1K27M virus ($n = 3$). **i** pSTAT3 Y705 protein expression in infected neurospheres as described in (**a**) ($n = 8$). Also see Supplementary Figure 1. All data are represented as mean with SEM, *$p < 0.05$, paired $t$ test

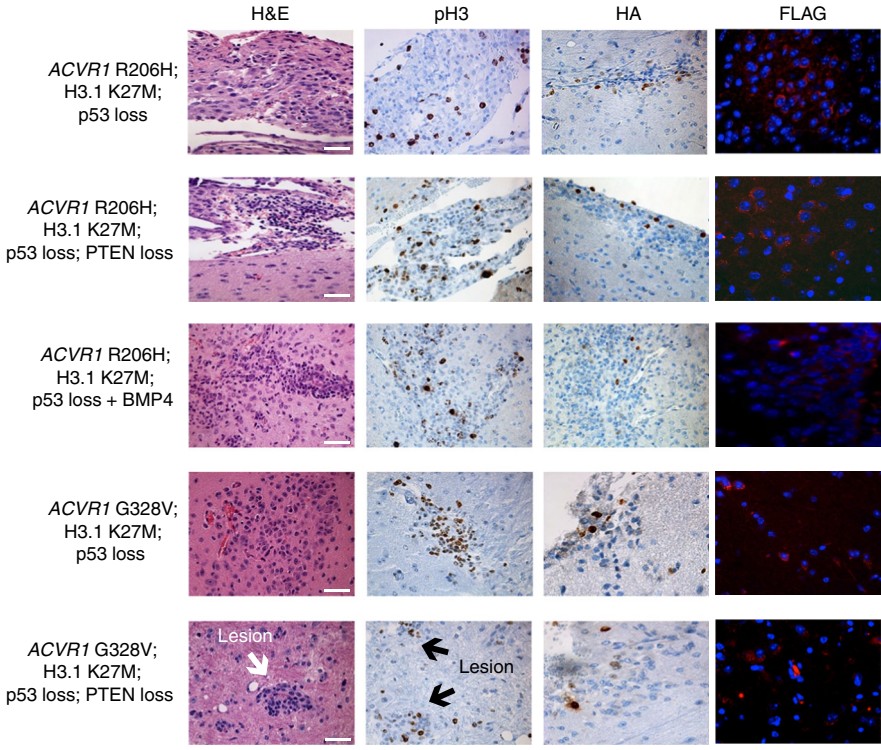

**Fig. 2** *ACVR1* mutants with H3.1K27M and p53 loss form glioma-like lesions. Representative H&E, IHC, and IF images for proliferation (anti-phospho-histone H3), H3.1K27M (HA), and *ACVR1* expression (FLAG) of glioma-like lesions for indicated groups (×40 magnification, scale bar = 50 µM). Also see Supplementary Table 1

H3.1K27M strengthens these mesenchymal transcriptomal changes.

**Formation of glioma-like lesions in mice**. To date, most studies examining the effects of *ACVR1* mutations have been performed in vitro while one study has shown that the expression of *ACVR1* mutants in p53 deficient astrocytes failed to induce tumor formation in vivo[15]. To determine whether *ACVR1* mutations could lead to tumorigenesis, nestin-expressing brainstem progenitors of neonatal Ntv-a;p53$^{fl/fl}$ mice were infected with either FLAG-epitope tagged *ACVR1* WT, R206H, G328V, or G328E and assessed for tumor formation. These mice did not develop tumors, regardless of which *ACVR1* mutation was expressed (Supplementary Table 1). However, when *ACVR1* R206H, G328V, or G328E was co-injected along with HA-epitope and FLAG-epitope tagged H3.1K27M and Cre to induce p53 loss, mice were able to generate glioma-like lesions in vivo

(4/10 = 40%, 4/13 = 31%, and 1/8 = 13%) (Fig. 2 and Supplementary Table 1). Additionally, while p53 loss was not sufficient to drive lesion formation (Supplementary Table 1), H3.1K27M and p53 loss together were sufficient for lesion formation (1/7 = 14%). Together, these results indicate that both H3.1K27M and p53 are required for glioma-like lesion formation.

Multiple virus combinations were tested in an effort to generate tumors with *ACVR1* mutations (Supplementary Table 1). However, in spite of our best efforts, tumors did not develop. As *ACVR1* mutations increase BMP signaling in vitro, we hypothesized that increased BMP signaling through the addition of exogenous BMP4 ligand or Activin-A, a recently identified ligand for *ACVR1* R206H in FOP[19], could lead to gliomagenesis. While exogenous BMP4 led to lesion formation in the context of *ACVR1* R206H, H3.1K27M, and p53 loss (3/6 = 50%), no tumors developed (Fig. 2 and Supplementary Table 1). Similarly, gliomagenesis did not occur with the addition of Activin A and surprisingly nor did lesion formation. Similarly, exogenous

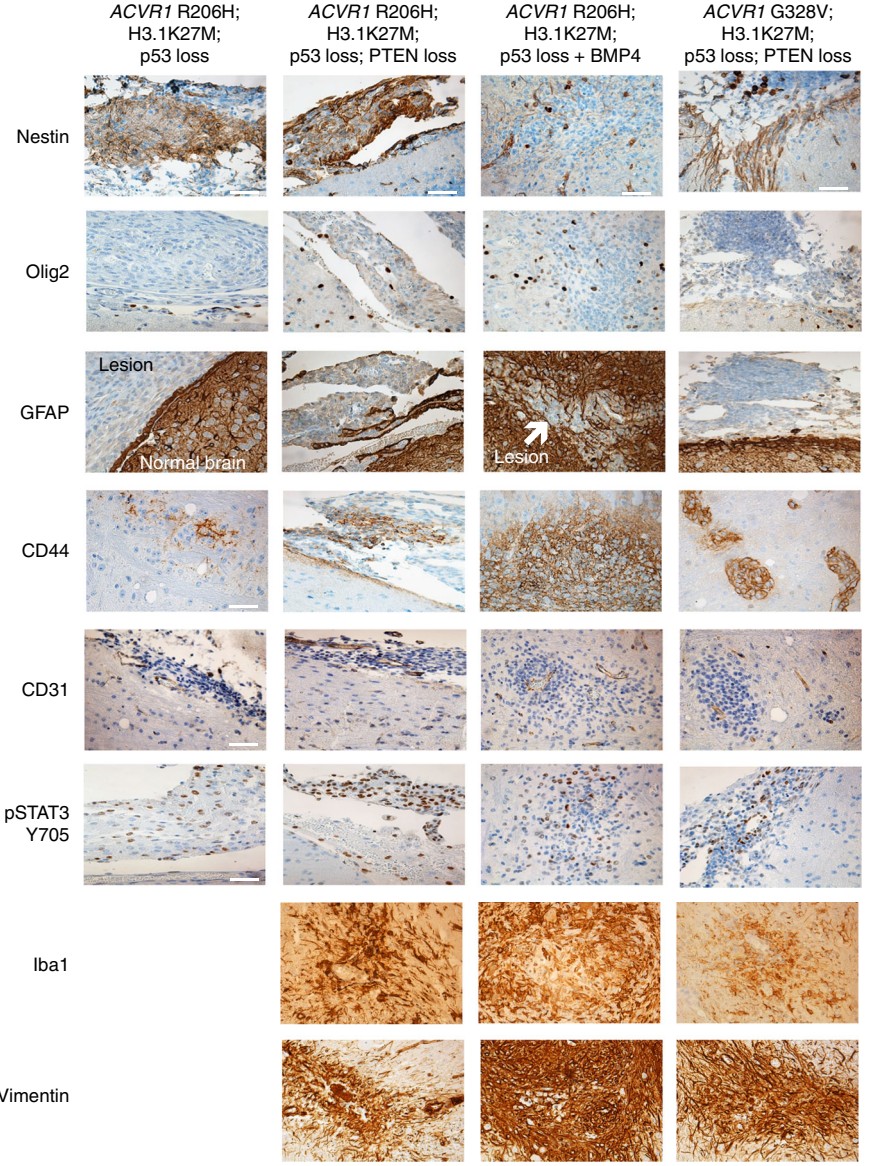

**Fig. 3** Glioma-like lesions express a mesenchymal-like phenotype. Representative IHC images of Nestin, Olig2, GFAP, CD44, CD31, pSTAT3 Y705, Iba1, and Vimentin of glioma-like lesions (×40 magnification, scale bar = 50 μM)

expression of Stat3 inhibited the development of these glioma-like lesions as well. Additionally, *ACVR1* mutations have been associated with activating mutations in the PI3K pathway in human DIPGs, including inactivating *pten* mutations or deletions[6,14]. Therefore, we sought to determine whether loss of PTEN could cooperate with *ACVR1* mutations, H3.1K27M, and p53 loss in Ntv-a;p53[fl/fl];PTEN[fl/fl] mice to form tumors. While the addition of PTEN loss led to the development of glioma-like lesions (6/10 = 60% for *ACVR1* R206H and 6/15 = 40% for *ACVR1* G328V), gliomagenesis did not occur (Fig. 2 and Supplementary Table 1). Immunohistochemistry staining of the glioma-like lesions demonstrated that cells within the lesions were proliferating and that a subset of cells expressed H3.1K27M and the mutant *ACVR1* (Fig. 2).

**Glioma-like lesions express a mesenchymal immunophenotype.** To further characterize the glioma-like lesions, all groups except *ACVR1* G328V, H3.1K27M, and Cre (because they were too small) were stained for Nestin, Olig2, and GFAP as these markers have been expressed in at least a subset of human DIPGs[20] (Fig. 3). While most cells within the lesions expressed Nestin, Olig2 staining was minimal. This absence of Olig2 staining suggested that these glioma-like lesions were not DIPGs as Olig2 is a robust marker of DIPG tumor cells[20,21]. However, expression profiling of DIPG biopsies has revealed that DIPGs with H3.1K27M and *ACVR1* mutations exhibit more of a mesenchymal phenotype with a pro-angiogenic signature[21,22]. In light of these observations and in combination with our RNASeq results, we then sought to determine whether the glioma-like lesions fit into the latter group. Lesions subsequently stained for endothelial marker CD31 and mesenchymal marker CD44 were found to express both (Fig. 3). It should be noted that although immunostaining of human DIPGs for CD44 has not been reported, CD44 mRNA has been reported to be expressed in a subset of human DIPGs by two independent groups, one of which noted that CD44 is expressed in DIPG associated microglia[22,23]. To further explore a possible mesenchymal phenotype and link to STAT3 signaling identified in our in vitro experiments[18], we immunostained for pSTAT3 and found it to be expressed

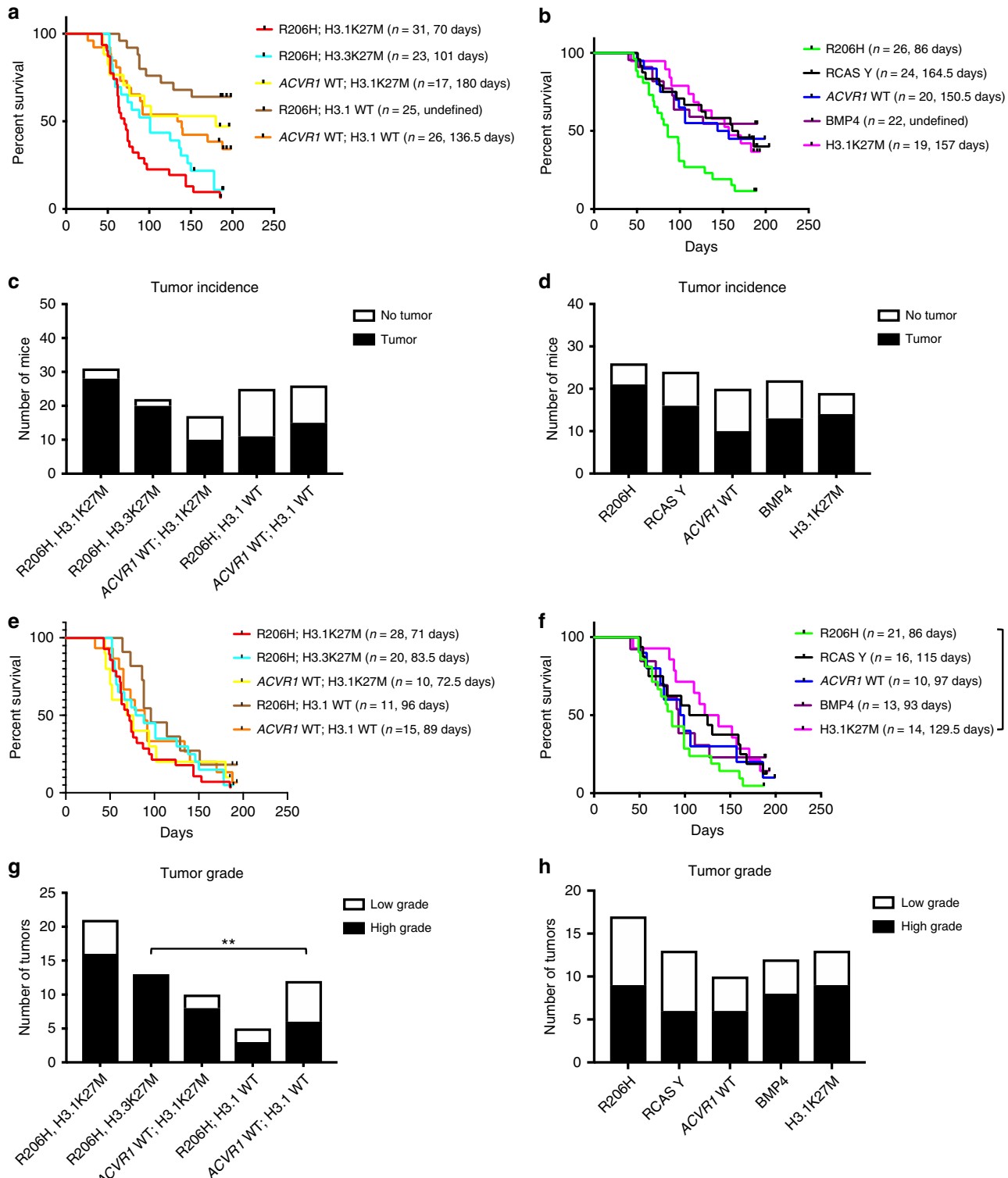

throughout the lesions as well (Fig. 3). However, due to the small size of the lesions we were unable to perform double immuno-fluorescence to determine if the cells expressing H3.1K27M and mutant *ACVR1* are also expressing pSTAT3. Additional mesenchymal markers vimentin and Iba1, which have been previously reported to stain human DIPGs[23,24], were also both expressed. Collectively, this data suggests that the glioma-like lesions display more of a mesenchymal phenotype rather than an oligodendroglial phenotype and is consistent with our observations in vitro.

**ACVR1 R206H and H3.1K27M reduce survival.** Because *ACVR1* mutations, even in combination with multiple additional mutations, were not sufficient to generate DIPGs in vivo, we incorporated the PDGFA ligand into our model as PDGFRA signaling is known to cooperate with H3.3K27M and p53 loss in other model systems[25,26] and as some human DIPGs with *ACVR1* mutations also had PDGFRA amplifications[6,15]. Mice infected with *ACVR1* R206H, H3.1K27M, Cre, and PDGFA had a median survival of approximately 70 days (Fig. 4a). This was significantly decreased compared to control mice that were

**Fig. 4** *ACVR1* R206H and H3.1K27M significantly decrease survival and increase tumor incidence. **a–h** All mice were injected with RCAS-PDGFA and RCAS-Cre along with additional RCAS viruses as indicated. **a** Kaplan–Meier survival curve of Nestin tv-a; p53$^{fl/fl}$ mice that were injected with RCAS-*ACVR1* R206H and RCAS-H3.1K27M ($n = 31$) or RCAS-*ACVR1* WT and RCAS-H3.1K27M ($n = 17$). For control purposes, mice were injected with RCAS-*ACVR1* R206H and RCAS-H3.3K27M ($n = 23$) or RCAS-*ACVR1* R206H and RCAS-H3.1 WT ($n = 25$) or RCAS-*ACVR1* WT and RCAS-H3.1 WT ($n = 26$). For significant differences among groups see Table 1. *$p < 0.05$, log-rank test. **b** Kaplan–Meier survival curve of Nestin tv-a p53$^{fl/fl}$ mice that were injected with RCAS-*ACVR1* R206H ($n = 26$), RCAS-Y ($n = 24$), RCAS-*ACVR1* WT ($n = 20$), RCAS-BMP4 ($n = 22$), or RCAS-H31.K27M ($n = 19$). For significant differences among groups see Table 1. **c, d** Mice from (**a, b**) were sacrificed when euthanasia endpoints were reached or 6 months post injection in the absence of symptoms. Tumor presence was assessed by H&E staining and confirmed by a blinded neuropathologist. For tumor incidence rates and significant differences among groups see Table 2. **e, f** Kaplan–Meier survival curve of tumor-bearing mice determined in (**c, d**). For significant differences among groups see Table 3. **g, h** Tumor grade was determined from mice in (**c, d**) as described. For significant differences among groups see Table 4. All Kaplan–Meier curves were analyzed using the log-rank test, *$p < 0.05$. All tumor incidence and tumor grade data were analyzed using Fischer's exact test, *$p < 0.05$

### Table 1 *ACVR1* R206H and H3.1K27M significantly decrease survival

| Virus combination | *p* value |
|---|---|
| *ACVR1* R206H; H3.1K27M vs *ACVR1* WT; H3.1K27M | 0.0048** |
| *ACVR1* R206H; H3.1K27M vs *ACVR1* R206H; H3.1 WT | <0.0001**** |
| *ACVR1* R206H; H3.1K27M vs *ACVR1* WT; H3.1 WT | 0.0038** |
| *ACVR1* R206H; H3.1K27M vs *ACVR1* WT | 0.0005*** |
| *ACVR1* R206H; H3.1K27M vs H3.1K27M | 0.0002*** |
| *ACVR1* R206H; H3.1K27M vs RCAS Y | 0.0002*** |
| *ACVR1* R206H; H3.1K27M vs BMP4 | 0.0001*** |
| *ACVR1* R206H; H3.3K27M vs *ACVR1* R206H; H3.1 WT | 0.0002*** |
| *ACVR1* R206H; H3.3K27M vs *ACVR1* WT | 0.0344* |
| *ACVR1* R206H; H3.3K27M vs H3.1K27M | 0.0163* |
| *ACVR1* R206H; H3.3K27M vs RCAS Y | 0.0183* |
| *ACVR1* R206H; H3.3K27M vs BMP4 | 0.0096** |
| *ACVR1* WT; H3.1K27M vs *ACVR1* R206H | 0.0391* |
| *ACVR1* R206H; H3.1 WT vs *ACVR1* WT; H3.1 WT | 0.0331* |
| *ACVR1* R206H; H3.1 WT vs *ACVR1* R206H | <0.0001**** |
| *ACVR1* R206H vs RCAS Y | 0.0058** |
| *ACVR1* R206H vs *ACVR1* WT | 0.0098** |
| *ACVR1* R206H vs BMP4 | 0.0037** |
| *ACVR1* R206H vs H3.1K27M | 0.0056** |

*$p < 0.05$; **$p < 0.01$; ***$p < 0.001$; ****$p < 0.0001$, log-rank test

### Table 2 *ACVR1* R206H and H3.1K27M significantly increase tumor incidence

| Virus combination | Tumor incidence (total tumors/total injected) |
|---|---|
| *ACVR1* R206H; H3.1K27M | 28/31 = 90% |
| *ACVR1* R206H; H3.3K27M | 20/22 = 91% |
| *ACVR1* WT; H3.1K27M | 10/17 = 59% |
| *ACVR1* R206H; H3.1 WT | 11/25 = 44% |
| *ACVR1* WT; H3.1 WT | 15/26 = 58% |
| *ACVR1* R206H | 21/26 = 81% |
| RCAS Y | 16/24 = 67% |
| *ACVR1* WT | 10/20 = 50% |
| BMP4 | 13/22 = 59% |
| H3.1K27M | 14/19 = 74% |
| | ***p* value** |
| *ACVR1* R206H; H3.1K27M vs *ACVR1* WT; H3.1K27M | 0.0220* |
| *ACVR1* R206H; H3.1K27M vs *ACVR1* R206H; H3.1 WT | 0.0003*** |
| *ACVR1* R206H; H3.1K27M vs *ACVR1* WT; H3.1 WT | 0.0059** |
| *ACVR1* R206H; H3.1K27M vs *ACVR1* WT | 0.0023** |
| *ACVR1* R206H; H3.1K27M vs RCAS Y | 0.0428* |
| *ACVR1* R206H; H3.1K27M vs BMP4 | 0.0171* |
| *ACVR1* R206H; H3.3K27M vs *ACVR1* WT; H3.1K27M | 0.0262* |
| *ACVR1* R206H; H3.3K27M vs *ACVR1* R206H; H3.1 WT | 0.0008*** |
| *ACVR1* R206H; H3.3K27M vs *ACVR1* WT; H3.1 WT | 0.0204* |
| *ACVR1* R206H; H3.3K27M vs *ACVR1* WT | 0.0055** |
| *ACVR1* R206H; H3.3K27M vs BMP4 | 0.0339* |
| *ACVR1* R206H; H3.1 WT vs *ACVR1* R206H | 0.0095** |

*$p < 0.05$; **$p < 0.01$; ***$p < 0.001$, Fisher's exact test

injected with *ACVR1* WT; H3.1K27M, Cre, and PDGFA (median survival 180 days, **$p = 0.0048$, log-rank test), *ACVR1* R206H, H3.1 WT, Cre, and PDGFA (median survival undefined, ****$p < 0.0001$, log-rank test), or those injected with *ACVR1* WT; H3.1 WT, Cre, and PDGFA (median survival 136.5 days, **$p = 0.0038$, log-rank test) (Fig. 4a, Table 1). To determine whether H3.1K27M is unique in its ability to cooperate with *ACVR1* R206H, we replaced H3.1K27M with H3.3K27M. Interestingly, there was no significant difference in survival between mice that were injected with *ACVR1* R206H, H3.3K27M, Cre, and PDGFA (median survival 101 days) and mice that were injected with *ACVR1* R206H, H3.1K27M, Cre, and PDGFA, suggesting that H3.3K27M can substitute for H3.1K27M in this context.

To determine whether *ACVR1* R206H or H3.1K27M was most responsible for the decrease in survival, mice were infected with either *ACVR1* R206H or H3.1K27M along with Cre and PDGFA. Those that were infected with *ACVR1* R206H had significantly decreased survival (median survival 86 days) as compared to those that were infected with RCAS-Y (median survival 164.5 days, **$p = 0.0058$, log-rank test) (Fig. 4b, Table 1), while those infected with H3.1K27M (median survival 157 days) had a similar median survival to the RCAS-Y group ($p = 0.9360$, log-rank test). This suggests that *ACVR1* R206H can accelerate gliomagenesis in the absence of H3.1K27M but not vice versa

(Fig. 4a, b). Furthermore, neurospheres were derived from *ACVR1* R206H, H3.1K27M, Cre, and PDGFA primary tumors and injected into Ntv-a; p53$^{fl/fl}$ mice. Not only did secondary tumors develop, but gliomagenesis was also greatly accelerated (Supplementary Figure 2).

While examining the effect of *ACVR1* mutations on survival, we noted that mice not infected with *ACVR1* R206H were more likely to survive to the end of the study and remain asymptomatic as compared to mice that were infected with *ACVR1* R206H. Therefore, we hypothesized that *ACVR1* R206H might have an effect on tumor incidence. In fact, mice that were infected with

**Table 3 ACVR1 R206H and H3.1K27M significantly decrease survival in tumors**

| Virus combination | p value |
|---|---|
| ACVR1 R206H; H3.1K27M vs RCAS Y | 0.0397* |
| ACVR1 R206H; H3.1K27M vs H3.1K27M | 0.0128* |
| ACVR1 R206H vs H3.1K27M | 0.0417* |

*p < 0.05, log-rank test

**Table 4 ACVR1 R206H and H3.3K27M significantly increase tumor grade**

| Virus combination | p value |
|---|---|
| ACVR1 R206H; H3.3K27M vs ACVR1 WT; H3.1 WT | 0.0052** |
| ACVR1 R206H; H3.3K27M vs ACVR1 R206H | 0.0021** |
| ACVR1 R206H; H3.3K27M vs RCAS Y | 0.0172* |
| ACVR1 R206H; H3.3K27M vs ACVR1 WT | 0.0307* |
| ACVR1 R206H; H3.3K27M vs BMP4 | 0.0172* |
| ACVR1 R206H; H3.3K27M vs H3.1K27M | 0.0237* |

*p < 0.05; **p < 0.01, Fisher's exact test

ACVR1 R206H, H3.1K27M, Cre, and PDGFA had increased tumor incidence (28/31 = 90%) as compared to control mice that were infected with ACVR1 WT, H3.1K27M, Cre, and PDGFA (10/17 = 59%, *p = 0.0220, Fisher's exact test) or ACVR1 WT, H3.1 WT, Cre, and PDGFA (15/26 = 58%, **p = 0.0059, Fisher's exact test) (Fig. 4c and Table 2). Surprisingly, mice that were infected with ACVR1 R206H, Cre, and PDGFA but not H3.1K27M or ACVR1 R206H, H3.1 WT, Cre, and PDGFA did not demonstrate an increase in tumor incidence (21/26 = 81% and 11/25 = 44%, respectively) compared to any group except each other (Fig. 4d and Table 2), suggesting that H3.1K27M is required for the effect of ACVR1 R206H on tumor initiation and is in agreement with our observations with the glioma-like lesions without PDGFA. Furthermore, H3.3K27M can substitute for H3.1K27M mutation and cooperate with ACVR1 R206H in tumor initiation as mice that were infected with ACVR1 R206H, H3.3K27M, Cre, and PDGFA also had a high tumor incidence (20/22 = 91%) similar to ACVR1 R206H, H3.1K27M, Cre, and PDGFA.

In light of this new information, survival curves were reanalyzed so that only tumor-bearing mice were examined in order to understand the effect of ACVR1 mutations on survival more clearly (Fig. 4e, f and Table 3). Interestingly, ACVR1 R206H; H3.1K27M; Cre; PDGFA tumor-bearing mice (median survival 71 days) had significantly decreased survival relative to only RCAS Y; Cre; PDGFA tumor-bearing mice (median survival 115 days *p = 0.0397, log-rank test) or H3.1K27M; Cre; PDGFA tumor-bearing mice (median survival 129.5 days *p = 0.0128, log-rank test). This effect appears to be driven by ACVR1 R206H as ACVR1 R206H; Cre; PDGFA tumor-bearing mice had decreased survival compared to H3.1K27M; Cre; PDGFA tumor-bearing mice (*p = 0.0417, log-rank test). Additionally, tumor grade was also examined by a blinded neuropathologist. Interestingly, the only significant difference in tumor grade among the different tumor cohorts was a subset of the comparisons with ACVR1 R206H; H3.3K27M; PDGF-A; p53 loss tumors. As they were 100% high-grade, tumor grade was significantly higher in this cohort relative to several other cohorts such as ACVR1 WT and H3.1 WT (Fig. 4g, h and Table 4). Collectively these results

suggest that together, ACVR1 R206H and H3.1K27M, significantly decrease survival and increase tumor incidence, suggesting a cooperating role for both mutations in tumor initiation.

**ACVR1 R206H and H3.1K27M upregulate BMP signaling and CD31.** To characterize the effect of ACVR1 R206H in vivo, immunohistochemistry staining was performed examining BMP pathway upregulation and angiogenesis as Id genes are known to regulate angiogenesis[27]. Mice that were infected with ACVR1 R206H, H3.1K27M, Cre, and PDGFA had increased staining for Id1 and Hes1, a component of the Notch pathway and also a target of the BMP signaling pathway[28]. It should be noted that while IHC for Hes1 has yet to be performed in human DIPGs, Hes1 has been reported to be expressed at both the mRNA and protein of levels of human DIPGs[29]. Together, these results indicated that BMP signaling was upregulated and also appeared to be driven by ACVR1 R206H (Fig. 5a). These tumors also demonstrated increased CD31 expression (Fig. 5a), suggesting a possible role for ACVR1 R206H in promoting angiogenesis. Staining for Nestin, Olig2, and GFAP revealed no significant differences among the different genotypes (Supplementary Figure 3). Western blot analysis of tumor lysates demonstrated similar results as those observed by IHC staining, with ACVR1 R206H, H3.1K27M; PDGF-A; p53 deficient mutant tumors having significantly higher protein levels of Id1, Hes1, and CD31 than other tumor types, with ACVR1 R206H contributing most to these higher levels (Fig. 5b–d). Phosphorylated STAT3 Y705 expression was also analyzed as ACVR1 R206H upregulated these levels in vitro. While not statistically, ACVR1 R206H, Cre, and PDGFA tumors lysates showed increased phosphorylated pSTAT3 levels as compared to RCAS Y, Cre, and PDGFA tumor lysates (Fig. 5e). Together, these results provide additional evidence that ACVR1 R206H promotes a mesenchymal signature in the presence of H3.1K27M.

**LDN212854 decreases proliferation in vitro and in vivo.** To validate our model and determine whether increased phosphorylated STAT3 Y705 levels are also observed in the human disease, human ACVR1 mutant lysates were analyzed for phosphorylated STAT3 levels. Indeed, ACVR1 mutant tumor tissue had significantly increased phosphorylated STAT3 Y705 levels as compared to their normal brain tissue counterparts (Fig. 6a). As both BMP and STAT3 signaling pathways appeared to be possible therapeutic targets for treating ACVR1 mutant DIPGs, mice were infected with ACVR1 R206H, Cre, PDGFA, and Noggin, an extracellular inhibitor of the BMP pathway, or STAT3 DN (Y705F) to inhibit the STAT3 pathway and assessed for survival benefit. Results showed that mice that were infected with Noggin had increased survival (median survival 132 days) as compared to mice infected with RCAS Y (median survival 83.5 days, *p = 0.0271, log-rank test). Mice that were infected with STAT3 DN showed no significant increase in survival (Fig. 6b). BMP and STAT3 signaling pathway inhibition was confirmed as Id1 and phosphorylated SMAD1/5/8 protein levels were decreased in mice injected with Noggin and phosphorylated STAT3 levels were decreased in mice injected with STAT3 DN (Fig. 6c). Interestingly, mice that were infected with STAT3 DN also had significantly decreased TNC and Vimentin levels as compared to mice infected with RCAS Y (Fig. 6d), demonstrating that mesenchymal gene upregulation by ACVR1 R206H is partly regulated by STAT3 signaling. Additionally, no significant decrease in tumor incidence or tumor grade was seen in STAT3 DN infected mice (Supplementary Figure 4). These results suggest that inhibition of the BMP pathway would be the more beneficial

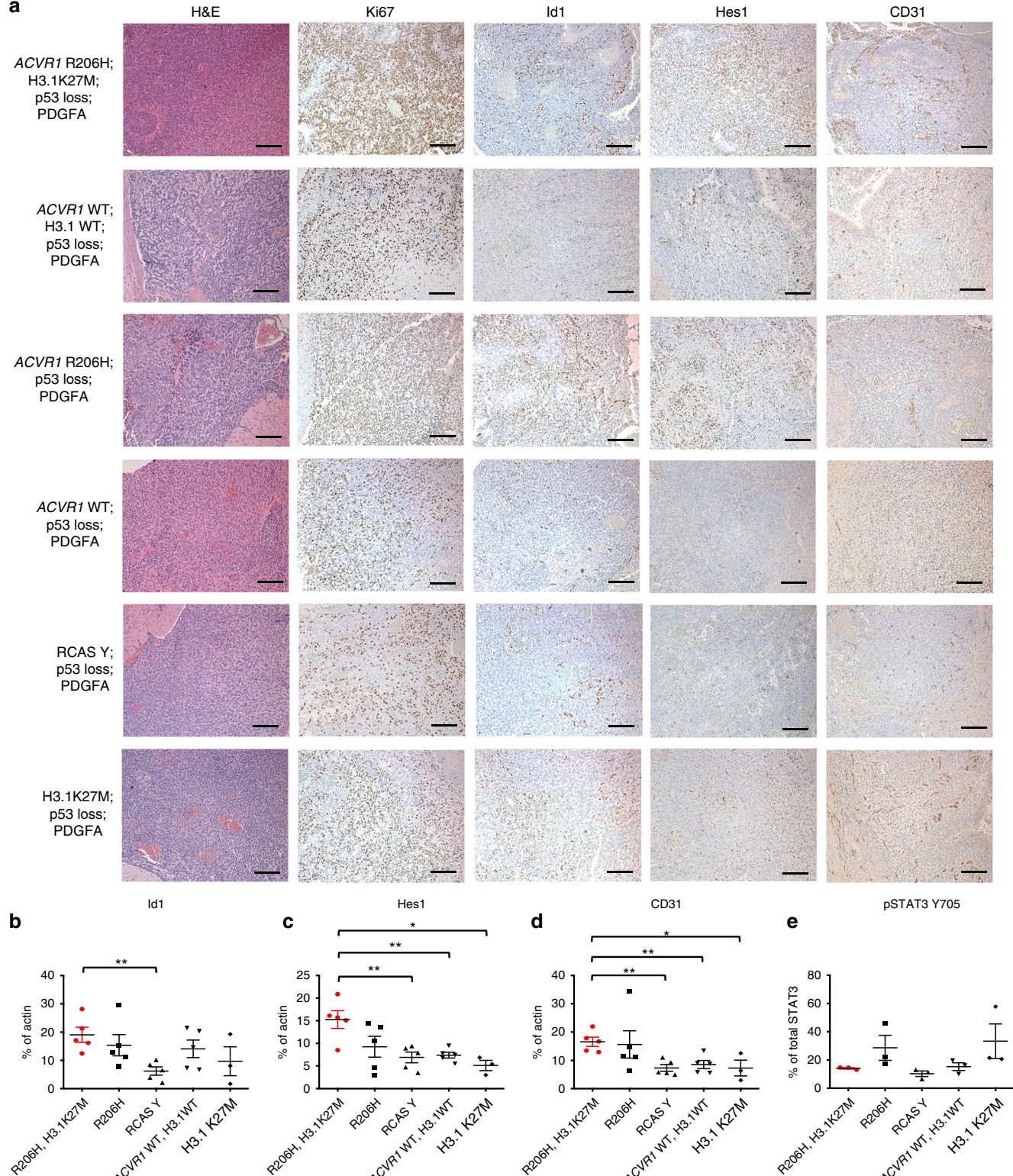

**Fig. 5** *ACVR1* R206H and H3.1K27M increase BMP signaling and CD31 expression in vivo. **a–e** All mice were injected with RCAS-PDGFA and RCAS-Cre along with additional RCAS viruses as indicated. **a** Representative H&E and IHC images of Ki67, Id1, Hes1, and CD31 in all injected groups. ×10 magnification, scale bar = 200 μM. **b–e** Western blot analysis of Id1 (**b**), Hes1 (**c**), CD31 (**d**), and pSTAT3 Y705 (**e**) expression from tumor-derived lysates from (**a**). For (**b–d**) all groups *n* = 5 except RCAS-PDGFA, RCAS-H3.1K27M, and RCAS-Cre injected group where *n* = 3. For (**e**) all groups *n* = 3. Data are represented as mean with SEM. *$p < 0.05$, unpaired *t* test

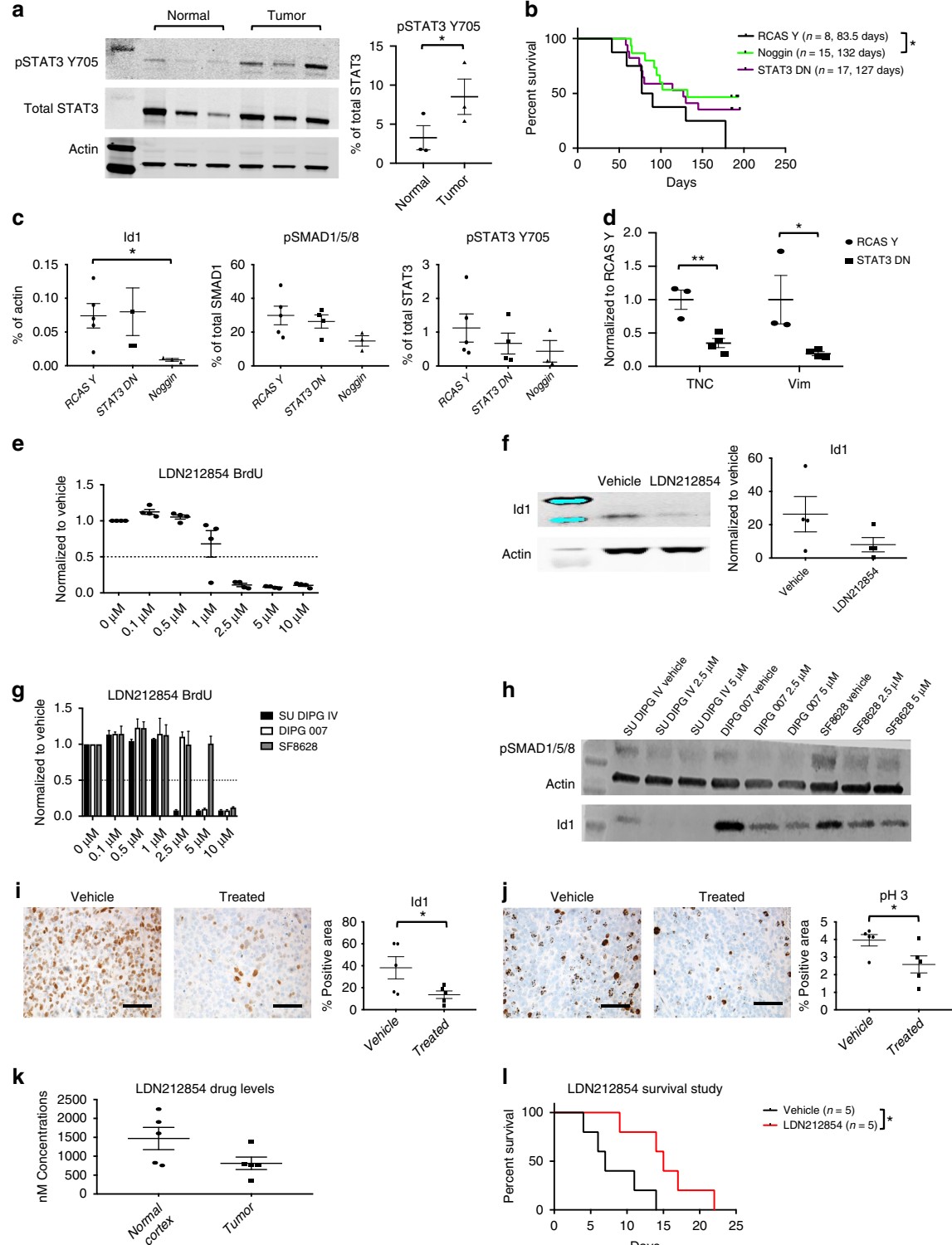

therapeutic strategy than targeting the STAT3 pathway for *ACVR1* mutant DIPGs.

Currently, no pharmaceutical agents exist that specifically target *ACVR1* R206H. Therefore, to determine whether *ACVR1* R206H could be used as a therapeutic target, neurospheres derived from *ACVR1* R206H, H3.1K27M, Cre, and PDGFA primary tumors were incubated with LDN212854, an *ACVR1* inhibitor that binds ALK-2 more potently than other ALK receptors[30]. LDN212854 treatment led to decreased proliferation

(Fig. 6e) and cell survival (Supplementary Figure 4) in all cell lines. However, it should be noted that LDN212854 had the same effect on neurospheres derived from primary tumors in mice not infected with *ACVR1* R206H (Supplementary Figure 4). Id1 expression was also decreased in LDN212854 treated neurospheres, indicating that LDN212854 was inhibiting the BMP pathway (Fig. 6f). To confirm our mouse studies, *ACVR1* mutant and wildtype human cell lines were treated with LDN212854. *ACVR1* mutant human cell lines SU DIPG IV (*ACVR1* G328V;

**Fig. 6** Noggin and *ACVR1* inhibitor LDN212854 increase survival in vivo. **a** pSTAT3 Y705 protein expression in normal brain tissue compared to tumor tissue from *ACVR1* mutant DIPG human samples ($n = 3$). *$p < 0.05$, paired $t$ test. **b** Kaplan–Meier survival curve of Nestin tv-a; p53$^{fl/fl}$ mice that were infected with RCAS-*ACVR1* R206H, RCAS-PDGFA, RCAS-Cre, and either RCAS-Y ($n = 8$), or RCAS-Noggin ($n = 15$), or RCAS-STAT3 DN ($n = 17$). *$p < 0.05$, log-rank test. **c** Id1, pSMAD1/5/8, and pSTAT3 Y705 protein expression from tumor-derived lysates of mice infected in (**b**) (RCAS-Y $n = 5$, RCAS-STAT3 DN $n = 4$, and RCAS-Noggin $n = 3$). *$p < 0.05$, unpaired $t$ test. **d** qRT-PCR analysis of TNC and Vimentin of RCAS-PDGFA, RCAS-Cre, RCAS-*ACVR1* R206H and RCAS-Y infected mice ($n = 3$) vs RCAS-PDGFA, RCAS-Cre, RCAS-*ACVR1* R206H and RCAS-STAT3 DN ($n = 4$) injected mice. *$p < 0.05$, unpaired $t$ test. **e, f** Tumor-derived neurospheres from Nestin tv-a; p53$^{fl/fl}$ mice infected with RCAS-PDGFA, RCAS-Cre, RCAS-*ACVR1* R206H, and RCAS-H3.1K27M were treated with *ACVR1* inhibitor LDN212854 and assessed for proliferation (**e**) and Id1 (**f**) expression ($n = 4$). For (**f**), paired $t$ test. Also see Supplementary Figure 4. **g, h** *ACVR1* mutant and wildtype human DIPG lines were treated with *ACVR1* inhibitor LDN212854 and assessed for proliferation (**g**) and pSMAD1/5/8 and Id1 (**h**) expression. Three independent experiments were performed. Also see Supplementary Figure 4 for quantification analysis. **i, j** Nestin tv-a; p53$^{fl/fl}$ mice were injected with RCAS-*ACVR1* R206H, RCAS-H3.1K27M, RCAS-PDGFA, RCAS-Cre, and RCAS-Luciferase. **i** Id1 and phospho-histone 3 (**j**) IHC staining in vehicle vs LDN212854 treated mice ($n = 5$ mice/group, ×40 magnification, scale bar = 50 μM). *$p < 0.05$, unpaired $t$ test. **k** LDN212854 drug levels measured in the cortex and tumor of drug-treated mice. **l** For survival studies, Nestin tv-a; p53$^{fl/fl}$ mice were injected with RCAS-*ACVR1* R206H, RCAS-H3.1K27M, RCAS-PDGFA, RCAS-Cre. Kaplan–Meier survival curve of vehicle vs LDN212854 treated mice. *$p < 0.05$, log-rank test. All data are represented as mean with SEM

H3.1K27M) and DIPG007 (*ACVR1* R206H; H3.3K27M) were more sensitive to LDN212854 treatment and had decreased proliferation at lower drug concentrations compared to *ACVR1* wildtype line SF8628 (Fig. 6g). All LDN212854 treated human lines showed significantly decreased phosphorylated SMAD1/5/8 and Id1 protein levels (Fig. 6h and Supplementary Figure 4), indicating inhibition of the BMP pathway. Mouse neurospheres and human cell lines were also treated with a second *ACVR1* inhibitor LDN214117, an inhibitor that is also a potent inhibitor of ALK2[31]. In mouse neurosphere lines, while treatment with LDN214117 did appear to inhibit the BMP pathway as evident by decreased Id1 expression, proliferation and cell viability levels were not significantly affected (Supplementary Figure 4). Similar results were also observed in LDN214117 treated human cell lines (Supplementary Figure 4). Together, these results suggest that inhibition of the BMP pathway may not be sufficient for a significant anti-tumor effect and that observed anti-tumor effects of LDN212854 may be mediated by off-target effects. Further studies will need to be done to confirm this.

However, because in vitro treatment with LDN212854 with mouse and human cell lines proved encouraging, we then sought to determine the efficacy of LDN212854 in vivo using *ACVR1* R206H, H3.1K27M, Cre, and PDGFA infected mice. Pharmacokinetics studies revealed that LDN212854 had good brain penetration and that LDN212854 treated mice had decreased Id1 expression and proliferation as compared to vehicle-treated mice (Fig. 6i–k). Survival studies of mice recipient of LDN212854 demonstrated a 2.1× increased survival with a median survival of 15 days compared to vehicle-treated mice with a median survival of 7 days after the beginning of treatment (Fig. 6l) (*$p = 0.0244$, log-rank test). Together, these results show that treatment with LDN212854 may be a worthwhile strategy to evaluate in a clinical trial for treating *ACVR1* mutant DIPG.

## Discussion

The role of BMP signaling in gliomagenesis is complex. Previous studies support the idea of BMP signaling as a tumor suppressor pathway in gliomas as it has been shown to block the proliferation of neural stem cells and glioma-initiating cells[32]. In contrast, it has also been shown that BMP signaling can also be tumor promoting in certain contexts[33,34]. Herein we study the effects of mutations in *ACVR1*, a serine threonine type I receptor in the BMP pathway, on DIPG biology. *ACVR1* has been implicated in oligodendroglial differentiation, but its role in the development of the pons is not clear[28]. To date, most identified *ACVR1* mutations have been restricted to DIPGs with the exception of one reported

case of a pediatric glioblastoma that arose in the spinal cord and harbored a G328E mutation[35].

To study the role of *ACVR1* in DIPG pathogenesis, brainstem progenitors were infected with three common *ACVR1* mutants: R206H, G328V, and G328E which resulted in differential effects on proliferation and cell survival with *ACVR1* R206H being the most potent mutation, *ACVR1* G328V being intermediate, and *ACVR1* G328E being the least potent. Surprisingly, the level of BMP pathway activation by the different *ACVR1* mutations in brainstem progenitors in vitro did not directly correlate with their effects on proliferation and cell survival nor did we observe an additive effect in combination with H3.1K27M. As *ACVR1* R206H had the largest effect on proliferation and cell survival in vitro, we performed RNAseq on brainstem progenitors infected with *ACVR1* R206H or *ACVR1* WT to identify target genes of this mutation. We observed significantly more differentially expressed genes between *ACVR1* R206H and *ACVR1* WT in the presence of H3.1K27M than without and noted that expression of *ACVR1* R206H in nestin-expressing brainstem progenitors upregulated genes that have been previously implicated in the mesenchymal subtype of glioma, namely CD44, Snail2, and Tenascin C[22,36,37]. Furthermore, GSEA analysis demonstrated that Stat3 signaling and EMT are positively enriched in the *ACVR1* R206H infected cells in comparison to *ACVR1* WT infected cells both with and without H3.1K27M. In support of the GSEA analysis, western blot analysis confirmed that brainstem progenitors infected with *ACVR1* R206H have significantly higher levels of Stat3 phosphorylation than *ACVR1* WT. As Stat3 signaling has been previously implicated in the mesenchymal transformation of adult glioblastomas, we hypothesized that the upregulation of mesenchymal markers by *ACVR1* R206H in the presence of H3.1K27M is at least in part due to the activation of Stat3 signaling. We provide some evidence in support of this by demonstrating that exogenous expression of a Stat3 dominant negative (Y705F) during tumor initiation with *ACVR1* R206H; PDGF-A; and p53 loss results in downregulation of two mesenchymal genes that were upregulated by *ACVR1* R206H in vitro, namely vimentin and tenascin C.

Through the development of our DIPG genetic model that incorporated *ACVR1* mutations, it noted that three of the most common *ACVR1* mutants (*ACVR1* R206H, *ACVR1* G328V, and *ACVR1* G328E) were not sufficient to induce DIPGs, even in combination with H3.1K27M, p53 loss, and PTEN loss. We did observe the formation of glioma-like lesions in a subset of these mice at 6 months post infection. Through these experiments, we discovered that the glioma-like lesions induced by mutant *ACVR1*, H3.1K27M, and p53 loss express a mesenchymal

phenotype based on the expression of CD44, CD31, pSTAT3, Iba1, vimentin, and minimal Olig2 expression. The original description of the mesenchymal glioma subtype in adult gliomas noted that mesenchymal gliomas harbor an astrocytic histology, which is consistent with our observations that a pathway that promotes astrocytic differentiation such as the BMP pathway can promote gliomagenesis of the mesenchymal subtype[38,39]. In addition, our observation of significantly increased CD31 immunostaining with the addition of ACVR1 R206H to the PDGF-A; p53 loss driven glioma model is also consistent with the mesenchymal subtype that is characterized by increased angiogenesis. Of note, an angiogenesis gene profile was also significantly enriched in the GSEA analysis of nestin-expressing brainstem progenitors infected with ACVR1 R206H relative to ACVR1 WT in the presence of H3.1K27M. Thus, our findings add ACVR1 to NF1 deletion as an additional genetic alteration found in human gliomas that promote mesenchymal transformation[40–43].

As mutant ACVR1, H3.1K27M, and p53 loss were not sufficient to induce murine DIPGs, PDGF-A was added to the model as PDGFRA amplifications are occasionally seen in human DIPGs that harbor ACVR1 mutations[6,15]. While the significance of the requirement for PDGF-A to develop ACVR1 mutant DIPGs with this modeling approach is unclear, PDGF-A has recently been reported to be expressed in at least a subset of DIPG tumor cells in at least 3 out of 4 different assays (cultured DIPG RNAseq, primary bulk DIPG RNAseq, primary single cell DIPG RNAseq)[23]. It is possible that tumor initiation in utero with these genetic alterations or in a different cell-of-origin postnatally may obviate the requirement for PDGF-A ligand (or other similar genetic alteration such as mutant PDGFRA). Nonetheless, ACVR1 R206H and H3.1K27M accelerated gliomagenesis and increased tumor incidence in the presence of PDGFA and p53 loss. Surprisingly, when we evaluated the effect of ACVR1 R206H and H3.1K27M separately from each other, ACVR1 R206H accelerated gliomagenesis without H3.1K27M while H3.1K27M was able to accelerate gliomagenesis only in the presence of ACVR1 R206H. These observations support our conclusions that the two genetic alterations cooperate in DIPG pathogenesis, and hence their strong association in the human tumors.

To determine whether targeting the BMP pathway is therapeutic in ACVR1 mutant DIPGs, we sought to inhibit BMP signaling by expressing exogenous Noggin, an extracellular antagonist of the BMP pathway, along with ACVR1 R206H, PDGFA, and p53 loss at tumor initiation. Noggin significantly inhibited the BMP pathway in vivo and successfully delayed gliomagenesis, suggesting that targeting the BMP pathway may be therapeutic. As our in vitro studies unraveled that Stat3 signaling is activated by ACVR1 R206H, we likewise expressed a dominant negative Stat3 instead of Noggin and observed that dominant negative Stat3 is not successful in significantly delaying gliomagenesis. Therefore, we proceeded to evaluate the anti-tumor effects of two ALK2 inhibitors that have been developed for FOP. LDN212854, was significantly more efficacious than LDN214117 in vitro, in both cell-lines derived from our murine model and human DIPG cell-lines. The reason for this is unclear and requires further investigation. Interestingly, LDN212854 has been demonstrated to significantly inhibit heterotopic ossification in multiple mouse model of FOP[17,30]. We noted that LDN212854 inhibited glioma neurosphere lines at a similar IC50s independent of expression of the ACVR1 R206H mutation. LDN214117 treatment did not reach an IC50 in the murine cell-lines nor in the human cell-lines even at a dose whereby Id1 levels were significantly inhibited. Together, these observations suggest that the efficacy of LDN212854 may be partly through an ALK2 independent mechanism and inhibition of ALK2 alone may not be sufficient for robust antitumor effect.

In conclusion, as BMP ligands are currently being used in clinical trials to treat high-grade gliomas in adults (ClinicalTrials. gov identifier NCT02869243)[44], it is important to understand how BMP signaling is contributing to gliomagenesis in certain contexts and inhibiting gliomagenesis in others. This may be a region-specific effect as ACVR1 mutations have only been observed thus far in the pons and spinal cord (i.e., midline gliomas). Our observations suggest that ACVR1 mutations have differing effects on proliferation and cell survival in vitro and ACVR1 mutations promote mesenchymal transformation in part by activating Stat3 signaling. Our genetic experiment with Noggin as well as evaluation of LDN212854 in mouse and human models suggest that inhibiting BMP signaling may be a successful therapeutic strategy in DIPGs that harbor ACVR1 mutations.

## Methods

**Mice.** Nestin-Tv-a (Ntv-a);p53[fl/fl] mice were generated by crossing Ntv-a mice with p53[fl/fl] mice[45]. Ntv-a;p53[fl/fl];PTEN[fl/fl] mice were generated by crossing Ntv-a;p53[fl/fl] mice with PTEN[fl/fl] mice. We have complied with all relevant ethical regulations for animal testing and research. All animals were used according to protocols approved at Duke University and Northwestern University Animal Care and Use Committee and the Guide for the Care and Use of Laboratory Animals (Animal Protocol A214-13-8 at Duke University and IS00005105 at Northwestern University).

**Infection of brainstem neurospheres with RCAS viruses.** Brainstem tissue was isolated from Ntv-a;p53[fl/fl] pups at postnatal day 3 (p3) and dissociated with papain and ovomucoid[46]. Cells were then cultured under neurosphere conditions in Dulbecco's Modified Eagle Medium (DMEM) supplemented with 10% proliferation supplement (Stem Cell Technologies), 1% Pen–Strep (Invitrogen), 20 ng/mL human basic FGF (Invitrogen), 10 ng/mL human EGF (Invitrogen), and 2 μg/mL heparin (Stem Cell Technologies) and incubated at 37 °C with 5% $CO_2$. Concentration of RCAS viruses was performed per the manufacturer's instructions (Clontech). Subsequently, neurospheres were then seeded in 96-well plates and infected with virus. Proliferation and cell survival assays were performed 3 days post infection with virus. Proliferation was measured by a bromodeoxyuridine (BrdU)-based cell proliferation ELISA assay kit (Roche) per the manufacturer's instructions and read on a Molecular Devices Versa Max tunable microplate. Cell viability was assessed through the use of a CellTiter-Glo Luminescent cell viability assay per the manufacturer's instructions and read on Turner Biosystems Modulus Microplate luminometer. All experiments were performed in triplicate wells.

**Western blot analysis.** Virus infected p3 brainstem progenitor neurospheres, primary tumor-derived neurospheres, or snap frozen tumors were made into lysates using RIPA buffer that contained 1× Protease Inhibitor Cocktail (Sigma Aldrich), 10 mM PMSF, 50 mM NaF, 1 mM NaVO$_4$, and 1 mM DTT[46]. Following protein quantification through BCA protein assays (Fisher Scientific) per the manufacturer's instructions, lysates were run on a NuPAGE 4–12% Bis–Tris gradient gel (Invitrogen) and transferred onto a nitrocellulose membrane. Membranes were incubated with primary antibodies at a 1:1000 dilution in Odyssey Blocking Buffer (Li-Cor) with 0.2% Tween-20 overnight at 4 °C. Incubation with secondary antibodies occurred at room temperature for 1 h at a dilution of 1:10,000. Antibodies used for Western blot analysis can be found in Supplementary Methods. All uncropped and unprocessed scans of representative blots can be found in Supplementary Data 3.

**qRT-PCR analysis.** Total RNA was isolated from virus infected p3 brainstem progenitor neurospheres or primary derived tumors through the use of an RNeasy kit (Qiagen) per the manufacturer's instructions. For qRT-PCR validation, cDNA was synthesized from total mRNA using Superscript II and OligodT primers (Invitrogen). All qRT-PCR experiments were performed using triplicate wells. qRT-PCR analysis was performed on a Bio-Rad iQ5 Multicolor Real-Time PCR Detection System. Primers for all tested genes can be found in the Supplementary Methods. Relative expression of target genes was analyzed using the $\Delta\Delta C_t$ method.

**RNAseq analysis.** Paired-end fastq files were imported into Galaxy[47], aligned to the mm10 genome using RNA-STAR, and aligned reads were counted using HTSeq-count with the Ensembl mm10 transcriptome GTF file as the feature file. The following HTSeq-count parameters were used: stranded = no, mode = union, minimum alignment quality = 10, map nonunique or ambiguous reads = none. HTSeq-count files were imported into R (https://www.r-project.org/), genes with <10 reads base mean were removed, and differential expression analysis was performed with the DESeq2 package[48] using the DESeqDataSetFromHTSeqCount function with default settings. DESeq2 analyses were run comparing ACVR1 wild-type to ACVR1 R206H mutant samples in the presence or absence of H3.1K27M mutation. GSEA pre-ranked analysis was run using genes ranked according to the

Wald statistic with the following parameters: permutations = 1000, enrichment statistic = classic, max size = 500, min size = 20, normalization mode = meandiv.

**Generation of glioma-like lesions and brainstem gliomas**. DF1 cells were cultured in DMEM with 10% fetal bovine serum (FBS)[45]. Transfection of DF1 cells with RCAS plasmids was performed through the use of X-TremeGENE 9 (Roche) per the manufacturer's instructions. For generation of glioma-like lesions and brainstem gliomas, $1 \times 10^5$ virus producing DF1 cells were injected intracranially into the brainstem of neonatal Ntv-a;p53[fl/fl] or Ntv-a;p53[fl/fl];PTEN[fl/fl] pups (postnatal days 3–5) in a 1 µL volume using a Hamilton syringe. Combination of viruses were injected at a 1:1, 1:1:1, 1:1:1:1, or a 1:1:1:1:1 ratio. Mice were monitored daily and euthanized with $CO_2$ when they were symptomatic (including an enlarged head, ataxia, weight loss up to 25%) or 6 months post injection if they were asymptomatic.

**Immunofluorescence**. Preneoplastic lesions were assessed for *ACVR1* expression by immunofluorescence staining. In brief, 5 µM thick sections were deparaffinized using xylene and decreasing concentrations of ethanol and rehydrated in PBS-T (0.1% Triton X-100). Following blocking with normal goat serum, slides were incubated with primary antibody (anti-DDK) at 4 °C overnight. Incubation with secondary antibody AlexaFluor-594 occurred at room temperature for 1 h. Slides were then mounted with Vectashield with Dapi (Vector Laboratories) and imaged using a Zeiss Axio Imager.

**Immunohistochemistry staining and analysis**. Tumor tissue was fixed in 10% formalin and embedded in paraffin by the Duke Pathology Core. 5 µm sections were then cut using a Leica RM2235 microtome. Immunohistochemistry analysis was performed using an automated processor (Discovery XT, Ventana Medical Systems, Inc.). Antibodies used for IHC analysis are listed in Supplementary Methods. For quantification of IHC staining, 10 different high-powered fields (×40 magnification) were captured for each tumor. Images were then analyzed using Metamorph software in which total nuclear area of positive staining was compared to total nuclear area.

**Tumor grading**. Formalin-fixed tumor samples were embedded in paraffin by the Duke Pathology Core and subsequently cut into 5 µM thick sections using a Leica RM2235 microtome. Hematoxylin and eosin (H&E) staining was performed using standard protocols. Tumor presence and grading were determined by a blinded neuropathologist. High-grade tumors were classified on the presence of vascular proliferation and/or pseudopalisading necrosis. Tumors with no vascular proliferation and/or pseudopalisading necrosis were classified as low grade. Tumors from found dead animals were excluded from grading analysis.

**Bioluminescence and MR imaging**. For in vivo bioluminescence imaging, Ntv-a; p53[fl/fl] mice were infected with RCAS-*ACVR1* R206H, RCAS-PDGFA, RCAS-H3.1K27M, RCAS-Cre, and RCAS-Luciferase. D-Luciferin was reconstituted per the manufacturer's protocol (Gold Biotechnology) and administered intraperitoneally (10 µL/g) following sedation with isoflurane. Images were collected until the peak signal was reached. All bioluminescence imaging was performed using an IVIS Lumina XR in vivo imaging system. MR imaging was conducted on a 7 Tesla Clinscan MRI (Bruker, Germany). Ten minutes prior to scanning each mouse was injected intraperitoneally with gadolinium-based MR contrast agent at 0.3 mmol/kg to allow for agent to reach the brain and enhance the tumor regions. Each mouse was then anesthetized using a mixture of isoflurane and 100% $O_2$ and placed in an MR compatible cradle. Respiration was monitored and recorded (SAI Instrument, NJ, USA) and body temperature was maintained at ~37 °C using a heated water system built-in the cradle. MR images were acquired using a dedicated four-channel mouse brain coil (Bruker, Germany) After initial localization sequences (tri-axial gradient echo sequences) a series of 2D images were acquired using T2 weighted Multi Spin Echo sequences in all 3 directions (transversal, longitudinal, and sagittal) with TR = 2000 ms and TE = 40 ms with a spatial resolution of ~80 µm and slice thickness of 0.7 mm. Finally, a 3D gradient T1 weighted echo sequence with isotropic resolution 150 µm was acquired with TR = 40 ms, TE = 3 ms, and Flip Angle FA = 10 yielding high-resolution 3D picture of the brain of each mouse. Post-acquisition inspection of the set of 2D and 3D images enabled assessment of morphological abnormalities associated with widespread tumor progression and detection of tumor masses.

**Treatment with *ACVR1* inhibitors LDN212854 and LDN214117**. For in vitro drug experiments, primary tumor-derived neurospheres were generated as described above. Neurospheres were then placed in 96-well plates and allowed to adhere for 24 h. Subsequently, cells were treated with LDN212854, LDN214117, or 0.1% DMSO for 24 h. BrdU, CellTiter-Glo, and Western blot assays were performed as described above to measure cell proliferation, cell viability, and pathway inhibition. For short term in vivo treatment with LDN212854 Ntv-a;p53[fl/fl] mice were infected with RCAS-*ACVR1* R206H, RCAS-PDGFA, RCAS-H3.1K27M, RCAS-Cre, and RCAS-Luciferase to facilitate the tracking of tumor development and growth over time. Tumor presence was confirmed through luminescence

imaging as described above. Mice bearing tumors were then randomized and treated with LDN212854 (10 mg/kg, twice daily, intraperitoneally) or vehicle (PBS) for 5 consecutive days. Tissue was harvested 4 h after the final dose and fixed in 10% formalin or snap frozen. For in vivo survival studies with LDN212854 Ntv-a; p53[fl/fl] mice were infected with RCAS-*ACVR1* R206H, RCAS-PDGFA, RCAS-H3.1K27M, and RCAS-Cre. Tumor presence was confirmed by MRI imaging as described above. Mice were then administered LDN212854 (6 mg/kg, intraperitoneally) or vehicle (PBS) twice a day until euthanasia endpoints were met as described above.

**Human DIPG models**. Primary pediatric human glioma cell lines, SF8628 and DIPG007 (HSJD-DIPG-007), containing H3.3K27M mutations and SU-DIPG-IV containing H3.1K27M mutations, were obtained from Dr. Rintaro Hashizume at Northwestern University (Chicago, IL), Dr. Michelle Monje at Stanford University (Stanford, CA), and Dr. Angel Montero Carcaboso at Hospital Sant Joan de Déu (Barcelona, Spain), in accordance with institutionally approved protocol at each institution[49–51]. SF8628 cells derived from surgical biopsy were maintained as an exponentially growing monolayer in complete medium consisting of DMEM (GIBCO 11965, Invitrogen) supplemented with 10% FBS with penicillin–streptomycin and plasmocin. SU-DIPG IV cell culture derived from DIPG autopsy tissue was grown as tumor neurospheres in tumor stem media (TSM) consisting of DMEM/F12 (Invitrogen), Neurobasal(-A) (Invitrogen), B27 (-A) (Invitrogen), human-bFGF (20 ng/mL; Shenandoah Biotechnology), human-EGF (20 ng/mL; Shenandoah Biotechnology), human PDGF-AB (20 ng/mL; Shenandoah Biotechnology), and heparin (10 ng/mL). DIPG007 (HSJD-DIPG-007) cells were derived from the autopsy and were maintained as an exponentially growing monolayer in TSM media supplemented with 5% FBS. Human cell cultures were validated by DNA fingerprinting using short tandem repeat (STR) analysis (PowerPlex16HS—Human specific; includes a mouse marker for detection of mouse DNA) and checked for mycoplasma contamination. No mycoplasma was detected. STR analysis can be found in Supplementary Data 4 and Supplementary Figures 5–7. Proliferation experiments using human DIPG cell lines was measured by a bromodeoxyuridine (BrdU) based cell proliferation ELISA assay kit (Roche) as described above. All experiments were performed in triplicate wells for a total of 3 independent experiments. Additional de-identified human DIPG lysates harboring *ACVR1* mutations were collected upon autopsy at Children's National Medical Center as part of an IRB-approved protocol, IRB-1339, PI Dr. Javad Nazarian and used in our study. For western blot analysis using human DIPG lysates, a total of 3 independent experiments were also performed.

**Pharmacokinetic analysis**. Snap frozen tissue was collected from Ntv-a;p53[fl/fl] mice injected with RCAS-*ACVR1* R206H, RCAS-PDGFA, RCAS-H3.1K27M, RCAS-Cre, and RCAS-Luciferase and treated with LDN212854 (10 mg/kg, twice daily, intraperitoneally) or vehicle (PBS) for 5 consecutive days. Measurement of drug levels including sample processing and liquid chromatography tandem-mass spectrometry (LC/MS/MS) assay was performed by the Duke Cancer Institute, Pharmaceutical Services-PK/PD Core Lab as described previously[52].

**Statistical analysis**. Statistical analysis was performed using GraphPad Prism software (Version 7.0). For p3 brainstem progenitor neurosphere in vitro assays, Western blots, and qRT-PCR experiments data are represented as the mean with SEM and analyzed using paired *t* tests (two-tailed). Sample sizes in figure legends represent the number of cell lines from independent p3 litters. All survival curves were analyzed by log-rank (Mantel–Cox) test. Tumor incidence and grade were measured by Fisher's exact test (two-tailed). Quantification of in vivo experiments including Western blot analysis from tumor-derived lysates, IHC staining, and tumor-derived qRT-PCR analysis were performed using unpaired *t* tests (two-tailed). For all tests, *p* values of less than 0.05 were considered significant.

**Reporting summary**. Further information on experimental design is available in the Nature Research Reporting Summary linked to this article.

## Data availability

RNAseq data that support the findings of this study have been deposited in Geo Bank under the accession number GSE125627. The authors declare that all other data supporting the findings of this study are available within the article and Supplementary Information files.

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

## Acknowledgements

The authors would like to thank A. Chung and K. Barton for their technical support. RCAS-Noggin was generously provided by Cliff Tabin. RCAS-BMP4 was generously provided by Arhat Abzhanov. RCAS-Stat3 was generously provided by Ganesh Rao. This work was supported by the Damon Runyon Cancer Research Foundation, the Stewart Trust Foundation, R01 CA197313, K02-NS086917, Madox's Warriors, Fly the Kite Foundation, Cristian Rivera Foundation, and John McNicholas Pediatric Brain Tumor Foundation.

## Author contributions

C.M.H. designed and performed the experiments, collected and analyzed the data, and wrote the manuscript. F.J.C., G.H., and K.M. helped with experiment design and data analysis. M.M.R. and H.J.C. helped with data collection. J.N. provided human DIPG samples. R.H. provided human DIPG cell-lines. Tumor grading was performed by R.M. P.Y. provided LDN212854 and LDN214117. D.P. performed MR Imaging and analysis. S.G. performed RNAseq and GSEA analysis. O.J.B. conceived the project, analyzed the data, and wrote the manuscript.

## Additional information

**Competing interests:** The authors declare no competing interests.

