## [Peer Review File · Nature Communications]

Reviewers' comments:

Reviewer #1 (Remarks to the Author):

Recent evidence has shown frequent co-occurrence of ACVR1 mutations with H3.1K27M in diffuse intrinsic pontine gliomas. The present manuscript describes experiments in which expression of mutant ACVR1 is combined with H3.1K27M and p53 deletion in murine nestin-positive brainstem progenitors. This genetic combination did not induce gliomas, but did promote increased proliferation of cells that expressed markers for the mesenchymal subtype of glioblastomas. Co-expression of PDGFA in the cells did induce gliomas, whose growth was modified by ALK inhibitors.

While modest effects on the growth of PDGFA induced tumors are noted, this is largely a negative study that does not clarify the mechanisms of tumorigenesis in the context of ACVR1 mutations with H3.1K27M. The described phenotypic markers are not clearly consistent with DIPGs, making it difficult to generalize the results to human tumors with these mutations.

Reviewer #2 (Remarks to the Author):

In this manuscript by Hoeman et al, the authors describe experiments using mouse models to unravel the importance of mutations in ACVR1 and H3.1 in a subset of diffuse intrinsic pontine gliomas DIPG. DIPG is a highly lethal CNS tumor of children and in desperate need of better understanding. One of the recent observations is that histone mutations are found in the majority of these tumors. And one of these recurrent mutations shows co-occurrence of mutations in the gene encoding ALK2 (ACVR1). This paper uses in vivo mouse models and cell culture experiments to unravel this connection.

The experiments are well done, and highly detailed. The topic has general interest. There are a few things that would strengthen the paper:

Is there documentation that LDN212854 gets into the brain?

Does the ACVR1 mutation do anything collaboratively with the other histone mutations? In cell culture? Looking for some evidence that the co-occurrence of this specific histone mutation makes sense.

On page 10 there are many statements of increasing of lesion size and incidence of formation. But with small numbers. And I suspect not statistically significant. Best to not overstate on limited data without statistics.

In the analysis of the tumors and "glioma-like lesions". Was co immunofluorescence done to verify that the actual tumor cells per se had the characteristics listed? Rather than stroma.

Could a more concrete delineation of actual DIPG from "glioma-like lesions" be stated. It seems that PDGF is needed for the actual cancer, whatever that means.

Was the state of H3.1 in the tumors that didn't get the RCAS version of H3.1 analyzed? Is there a selection for the parallel alterations in the mouse version of this protein? Or was the AKT pathway activity analyzed in the tumors that didn't get the mutation in PI3K? again, maybe elevated activity was selected for when the tumor finally did arise.

The paragraph on page 14 is a bit confusing, as is the one on page 15. Maybe a summary of the results could replace these paragraphs and the detailed text here could be moved to

supplementary data.

On page 18 there seems to be 5 vectors all mixed together and infected mice. And one of them is luciferase. How sure are the authors that all 5 vectors got into the tumor cells, especially since luciferase is immunogenic when transferred. Germline luciferase when activated with Cre is less so I understand.

Reviewers' comments:

We thank the reviewers for their time and their constructive feedback. We have spent the last 4 months performing additional experiments both *in vitro* and *in vivo* to address the reviewers' comments and feel that the manuscript is now significantly improved from the original submission. We hope that the additional new data provided is sufficient to satisfy the reviewers' initial critiques. Reviewers' comments are listed below in italics followed by our responses.

Reviewer #1 (Remarks to the Author):

Recent evidence has shown frequent co-occurrence of ACVR1 mutations with H3.1K27M in diffuse intrinsic pontine gliomas. The present manuscript describes experiments in which expression of mutant ACVR1 is combined with H3.1K27M and p53 deletion in murine nestin-positive brainstem progenitors. This genetic combination did not induce gliomas, but did promote increased proliferation of cells that expressed markers for the mesenchymal subtype of glioblastomas. Co-expression of PDGFA in the cells did induce gliomas, whose growth was modified by ALK inhibitors.

While modest effects on the growth of PDGFA induced tumors are noted, this is largely a negative study that does not clarify the mechanisms of tumorigenesis in the context of ACVR1 mutations with H3.1K27M.

Author's response- Thank you for your feedback. While we were surprised that expression of H3.1K27M, ACVR1 R206H, and p53 deletion in nestin progenitors of the neonatal brainstem were not sufficient to drive full DIPG pathogenesis, our observations that these three genetic alterations can induce cell proliferation in the neonatal brainstem as well as expression of markers for the mesenchymal subtype of glioblastomas are important for our understanding of DIPG pathogenesis. These results will be informative to the field as we need to understand why expression of these three genetic alterations is not sufficient for full gliomagenesis with this modeling approach. Further studies will be needed to determine if these three genetic alterations will be sufficient for full gliomagenesis if initiated *in utero* or in a different cell-of-origin postnatally.

It is worth noting that similar results have been observed with the more common histone mutation, H3.3K27M. So far only one group has observed that midline gliomas can arise with the combination of just H3.3K27M and p53 loss¹. In that experimental system, tumorigenesis was initiated using *in utero* electroporation of piggyback transposon-based vectors, and it took 6-8 months for tumors to develop. Several other groups, including our lab, have reported that PDGF signaling is required to induce gliomas with H3.3K27M^{2,3}. We are aware of several other groups that are working on developing genetic mouse models of DIPG using different technologies and so, with time, results from different experimental systems will allow us to better understand the role of each of these genetic alterations in DIPG pathogenesis.

Our results strongly suggest that H3.1K27M and *ACVR1* R206H are most important for tumor initiation. This is an important and novel finding with therapeutic implications. We also note that H3.1K27M and *ACVR1* R206H promote tumor progression. Future studies will be required to rigorously investigate whether these mutations are required for tumor maintenance.

To further clarify the mechanism of tumorigenesis in the context of *ACVR1* mutations and H3.1K27M, we performed GSEA analysis of differentially expressed genes identified in the *in vitro* RNAseq, where we infected nestin-expressing brainstem progenitors with either *ACVR1* R206H or *ACVR1* WT with and without H3.1K27M. Please note that we have re-analyzed the RNAseq by mapping to mm10, the more updated mouse genome, and so the differentially expressed gene lists are slightly different from those in the initial submission. In the re-analysis, there were only 24 significantly differentially expressed genes between *ACVR1* R206H and *ACVR1* WT in the absence of H3.1K27M, and 2,478 significantly differentially expressed genes between *ACVR1* R206H and *ACVR1* WT in the presence of H3.1K27M. GSEA analysis identified several important signaling pathways that were positively and negatively enriched in *ACVR1* R206H samples. A table listing the positively enriched gene-sets in the presence of H3.1K27M is included below as an example. The full results are in the revised manuscript (Figure 1f, h, Supplemental Figure 1e, g, and Supplemental Table 2 in the revised manuscript). Of note, epithelial mesenchymal transition (EMT) and Stat3 signaling were significantly enriched both with and without H3.1K27M (GSEAs in the presence of H3.1K27M are provided below as an example) suggesting that *ACVR1* R206H is primarily promoting these signatures, which is in line with our observations that mesenchymal markers are expressed in the glioma-like lesions induced by H3.1K27M, *ACVR1* R206H, and p53 deletion.

Positively enriched in R206H K27M vs ACVR1 wildtype K27M						
NAME	SIZE	ES	NES	NOM p-val	FDR q-val	FWER p-val
HALLMARK_HYPOXIA	177	0.284605	4.506472	0	0	0
HALLMARK_INTERFERON_GAMMA_RESPONSE	165	0.285716	4.281605	0	0	0
HALLMARK_TNFA_SIGNALING_VIA_NFKB	177	0.267777	4.121865	0	0	0
HALLMARK_MYOGENESIS	178	0.264587	4.055461	0	0	0
HALLMARK_INTERFERON_ALPHA_RESPONSE	81	0.384947	4.05023	0	0	0
HALLMARK_APOPTOSIS	145	0.257809	3.749518	0	0	0
HALLMARK_EPITHELIAL_MESENCHYMAL_TRANSITION	184	0.229131	3.618813	0	0	0
HALLMARK_IL2_STAT5_SIGNALING	166	0.215357	3.250515	0	0	0
HALLMARK_P53_PATHWAY	187	0.195365	3.101531	0	0	0
HALLMARK_ESTROGEN_RESPONSE_LATE	173	0.197221	3.047195	0	0	0
HALLMARK_CHOLESTEROL_HOMEOSTASIS	66	0.317097	3.040387	0	0	0
HALLMARK_IL6_JAK_STAT3_SIGNALING	67	0.300556	2.93493	0	0	0
HALLMARK_UV_RESPONSE_DN	138	0.216939	2.882152	0	0	0
HALLMARK_KRAS_SIGNALING_UP	170	0.168966	2.572891	0	1.84E-04	0.002
HALLMARK_ANGIOGENESIS	31	0.388261	2.578153	0.001942	1.97E-04	0.002
HALLMARK_ANDROGEN_RESPONSE	91	0.207544	2.372203	0	6.30E-04	0.007
HALLMARK_UV_RESPONSE_UP	143	0.160572	2.215355	0	0.002778	0.034
HALLMARK_INFLAMMATORY_RESPONSE	147	0.150565	2.179907	0	0.003139	0.04
HALLMARK_COAGULATION	105	0.176264	2.141786	0.002049	0.00398	0.054
HALLMARK_TGF_BETA_SIGNALING	53	0.238659	2.106529	0	0.004728	0.065
HALLMARK_APICAL_SURFACE	40	0.274568	2.063283	0.008097	0.005932	0.086
HALLMARK_GLYCOLYSIS	182	0.12905	2.029625	0.004057	0.006784	0.107
HALLMARK_ESTROGEN_RESPONSE_EARLY	180	0.130574	2.030223	0.00404	0.007035	0.106
HALLMARK_APICAL_JUNCTION	177	0.128009	1.977873	0.007828	0.009259	0.151
HALLMARK_WNT_BETA_CATENIN_SIGNALING	38	0.255967	1.944431	0.007648	0.010765	0.182
HALLMARK_HEDGEHOG_SIGNALING	35	0.255722	1.842112	0.005859	0.017032	0.273
HALLMARK_XENOBIOTIC_METABOLISM	160	0.097897	1.453075	0.093306	0.109005	0.892
HALLMARK_PROTEIN_SECRETION	91	0.120434	1.350297	0.142315	0.164366	0.964
HALLMARK_HEME_METABOLISM	166	0.089711	1.32364	0.151639	0.175214	0.979
HALLMARK_KRAS_SIGNALING_DN	127	0.095863	1.284426	0.169291	0.197597	0.989

Table

Legend- Part of Supplemental Table 2 depicting positively enriched gene-sets in

ACVR1 R206H/H3.1K27M vs. *ACVR1* WT/H3.1K27M. Negatively enriched gene-sets are included in Supplemental Table 2.

Figure Legend- (G) GSEA enrichment plots of IL-6_JAK_STAT3 signaling and Epithelial Mesenchymal Transition (EMT) gene-set identified by RNA-Seq analysis

With RT-PCR, we validated a subset of mesenchymal genes identified in the RNAseq analysis to be upregulated by the *ACVR1* R206H and investigated whether the same genes are upregulated in nestin-expressing brainstem progenitors infected with *ACVR1* G328V relative to *ACVR1* WT in the presence of H3.1K27M. Vimentin, tenascin C, CD44, and Snail 2 were all upregulated with the latter 3 being significantly upregulated by at least one of the *ACVR1* mutations. In line with these observations, the transcript levels of Sox10, a proneural marker, were reduced by the two *ACVR1* mutations (Please see below and in Figure 1g of the revised manuscript). For this manuscript, we decided to focus on the Stat3 signaling pathway because Stat3 signaling has been implicated in the mesenchymal subtype of glioblastomas⁴ and we observed upregulation of mesenchymal markers in both the *in vitro* and *in vivo* experiments in the absence of PDGF-A (glioma-like lesion). However, these mutations likely promote tumorigenesis through other mechanisms as well, which will be addressed in future studies.

Figure legend- qRT-PCR validation of select genes from neurospheres infected with *ACVR1* WT, R206H, or G328V and H3.1K27M virus (n = 3). Data are represented as mean with SEM, * p < 0.05, paired t test.

We confirmed the results from the GSEA analysis by performing western blots for Stat3 phosphorylation (Y705) of nestin-expressing brainstem progenitors infected with each of the three of the *ACVR1* mutants (R206H, G328V, and G328E) or *ACVR1* WT with or without infection with H3.1K27M and observed that all three *ACVR1* mutations increase Stat3 signaling, and *ACVR1* R206H significantly increasing Stat3 phosphorylation relative to *ACVR1* WT (see western blots below and Figure 1i in the revised manuscript). The addition of H3.1K27M to each of the *ACVR1* mutations or *ACVR1* WT did not significantly affect phosphorylated Stat3 levels. This is described at the top of page 10 of the revised manuscript and is pasted below:

Page 10- “Because the STAT3 signaling pathway has been shown to regulate mesenchymal genes in brain tumors⁴, we examined whether STAT3 signaling was activated by *ACVR1* mutations at the protein level. Indeed, cells that were infected with *ACVR1* mutations had modestly increased levels of phosphorylated STAT3 Y705 as compared to *ACVR1* WT infected cells, with *ACVR1* R206H demonstrating significantly increased levels (Fig. 1I). However, infection with both *ACVR1* mutations and H3.1K27M did not significantly increase phosphorylated STAT3 Y705 expression compared to infection with *ACVR1* mutations alone (Supplementary Fig. 1H). Together, these results indicate that one way *ACVR1* mutations along with H3.1K27M co-operate is by driving a mesenchymal profile which may be due in part to increased STAT3 signaling.”

Figure Legend- Western blot analysis of pSTAT3 Y705 expression in infected neurospheres as described in (A) (n = 8). Data are represented as mean with SEM, * p < 0.05, paired t test.

We also performed western blots using tumor lysates from the *in vivo* murine tumors with and without *ACVR1* R206H, H3.1K27M alone or in combination, and from the wildtype controls. Quantification of the western blot analysis for pSTAT3 Y705 is provided below and in Figure 5E in the revised manuscript. While not statistically significant, *ACVR1* R206H, Cre, and PDGFA tumor lysates showed increased phosphorylated pSTAT3 levels as compared to RCAS Y, Cre, and PDGFA tumor lysates. Revised text is below and in the revised manuscript on page 18:

Page 18- “Additionally, tumor lysates were also used to assess phosphorylated STAT3 Y705 expression as *ACVR1* R206H upregulated phosphorylated pSTAT3 Y705 levels *in vitro*. While not statistically significant, *ACVR1* R206H, Cre, and PDGFA tumor lysates showed increased phosphorylated pSTAT3 levels as compared to RCAS Y, Cre, and PDGFA tumor lysates (Fig. 5e).”

Figure Legend- Quantification of western blot analysis for pSTAT3 Y705 expression from tumor derived from lysates (n = 3 per tumor genotype). Data are represented as mean with SEM.

Next, to validate our findings with the mouse modeling in the human disease, we examined the levels of phosphorylated Stat3 in human DIPGs with *ACVR1* mutations and adjacent matched normal control brains. Below is the western blot (Figure 6A of the revised manuscript) and a graph of the quantification showing significantly increased

phosphorylated Stat3 Y705 relative to total Stat3 in human DIPG tumors with *ACVR1* mutations.

Figure legend- Western blot analysis of pSTAT3 Y705 expression in normal brain tissue compared to tumor tissue from *ACVR1* mutant DIPG human samples (n = 3). Data are represented as mean with SEM, * p < 0.05, paired t test.

Revised text from pages 18/19 is pasted below:

Pages 18/19- “In an effort to validate our observations using mouse modeling and determine whether this increase in phosphorylated STAT3 Y705 levels are also observed in the human disease, three human *ACVR1* mutant lysates were analyzed for phosphorylated STAT3 levels. Indeed, *ACVR1* mutant tumor tissue had significantly increased phosphorylated STAT3 Y705 levels as compared to their normal brain tissue counterparts (Fig. 6a) thereby further confirming our results”

Next, to evaluate the functional impact of Stat3 signaling *in vivo*, we generated an RCAS vector expressing a dominant negative (DN) Stat3 (Y705F) to determine whether expression of Stat3 DN will delay gliomagenesis in the presence of *ACVR1* R206H. Interestingly, expression of Stat3 DN did not significantly impact gliomagenesis in contrast to noggin, an extracellular inhibitor of BMP signaling that did significantly delay gliomagenesis. This is illustrated below and in Figure 6B of the revised manuscript. We also performed western blots to assess pathway inhibition-this is also illustrated below and in Figure 6C of the revised manuscript. Noggin significantly inhibited the BMP pathway as measured by ID1. Interestingly, phosphorylated Stat3 levels were reduced in both the Noggin and Stat3 DN cohorts, consistent with BMP signaling being upstream of Stat3 signaling. This justifies our focus on targeting the BMP pathway pharmacologically in this manuscript (the revised text is pasted below the two figure panels).

Figure Legend- Kaplan-Meier survival curve of Nestin^{tv-a}; p53^{fl/fl} mice that were injected with RCAS-PDGFA, RCAS-Cre, RCAS-ACVR1 R206H and RCAS-Y (n = 8), RCAS-Noggin (n = 15), or RCAS-STAT3 DN (n = 17). * p < 0.05, log-rank test.

Figure Legend- Western blot analysis of Id1, pSMAD1/5/8, and pSTAT3 Y705 expression from tumor derived lysates of mice injected in (B) (RCAS-Y n = 5, RCAS-STAT3 DN n = 4, and RCAS-Noggin n = 3). Data are represented as mean with SEM. * p < 0.05, unpaired t test.

Page 19- “As both the BMP and STAT3 signaling pathways appeared to be possible therapeutic targets for treating *ACVR1* mutant DIPGs, mice were injected with *ACVR1* R206H, Cre, PDGFA, and Noggin, an extracellular inhibitor of the BMP pathway, or STAT3 DN (Y705F) to inhibit the STAT3 pathway and assessed for survival benefit. Results showed that mice that were infected with Noggin had increased survival (median survival 132 days) as compared to mice infected with empty vector (median survival 83.5 days, * p = .0271) while mice that were infected with STAT3 DN showed no significant increase in survival (Fig. 6b). To confirm that the BMP and STAT3 signaling pathways were inhibited in infected mice, tumor derived lysates were used to assess Id1, phosphorylated SMAD1/5/8, and phosphorylated STAT3 levels. Indeed, Id1 and phosphorylated SMAD1/5/8 expression were decreased in mice injected with Noggin and phosphorylated STAT3 levels were decreased in mice injected with STAT3 DN (Fig. 6c).”

As we hypothesized that Stat3 signaling mediates at least a part of the mesenchymal profile induced by *ACVR1* R206H, we performed RT-PCR on *in vivo* tumors induced with *ACVR1* R206H; PDGF-A; p53 deficient and *ACVR1* R206H A; PDGF-A; p53 deficient; Stat3 Y705F and observed that a couple of the mesenchymal genes (vimentin and tenascin C) that we identified to be upregulated by the *ACVR1* mutations in the *in vitro* progenitor experiments are downregulated in the Stat3 Y705F tumors relative to the controls, suggesting that Stat3 signaling partly regulates the expression of mesenchymal markers induced by *ACVR1* mutations in the presence of H3.1K27M. This is illustrated below and in the revised manuscript, Figure 6D. The revised text is pasted below the figure panel.

Figure Legend- qRT-PCR analysis of TNC and Vimentin of RCAS-PDGFA, RCAS-Cre, RCAS-ACVR1 R206H and RCAS-Y injected mice (black circles, n = 3) compared to RCAS-PDGFA, RCAS-Cre, RCAS-ACVR1 R206H and RCAS-STAT3 DN (black squares, n = 4) injected mice. Data are represented as mean with SEM, * p < 0.05, unpaired t test.

Page 19- “Interestingly, mice that were injected with STAT3 DN also had significantly decreased TNC and Vimentin levels as measured by qRT-PCR compared to mice injected with RCAS Y (Fig. 6d), demonstrating that at least some of the mesenchymal genes upregulated by *ACVR1* R206H are in part regulated by STAT3 signaling.”

In conclusion, the revised manuscript demonstrates that part of the mechanism by which *ACVR1* mutations contribute to DIPG pathogenesis is by increasing Stat3 signaling, which in turn promotes the expression of mesenchymal markers such as CD44, Snail2 and Tenascin C. A couple of studies have compared H3.1K27M human DIPGs and H3.3K27M human DIPGs - Castel et al. 2015⁵ and 2018⁶- noting that H3.1-K27M mutated tumors exhibit a mesenchymal/astrocytic phenotype and a pro-angiogenic/hypoxic signature and are distinct from H3.3K27M mutant tumors, which are more proneural or oligodendroglial. Remarkably, our study uncovers that mutant *ACVR1* is important in driving this phenotype. The significant GSEA pathways upregulated by *ACVR1* R206H relative to *ACVR1* WT with and without H3.1K27M included epithelial to mesenchymal transition, Stat3 signaling, hypoxia, and angiogenesis. Our study helps unravel the complex signaling pathways downstream of mutant *ACVR1* and H3.1K27M and helps clarify their mechanism that is distinct from H3.3K27M.

To further strengthen the connection of our findings to the human disease, we performed studies in three human DIPG cell-lines (DIPG IV harbors *ACVR1* G328V and H3.1K27M mutations and was generated by Michelle Monje at Stanford University. DIPG007 has *ACVR1* R206H and H3.3K27M mutations and was generated by Dr. Carcaboso in Barcelona, Spain. The 3rd line DIPG SF8628 has the H3.3K27M but not the *ACVR1* mutation). All three human DIPG cell-lines showed decreased proliferation with LDN212854 treatment but the two *ACVR1* mutant lines were more sensitive to LDN212854 than the *ACVR1* WT line. Western blot analysis demonstrated that LDN212854 reduces SMAD 1/5/8 phosphorylation and ID1 levels across all three cell-lines (See figure below and Figure 6G and H in the revised manuscript and Supplemental Figure 4F & G). The revised text is pasted below the figure legend.

Figure Legend- *ACVR1* mutant and wildtype human DIPG lines were treated with *ACVR1* inhibitor LDN212854 for 24 hours and assessed for proliferation (G) and pSMAD1/5/8 and Id1 (H) expression by Western blot analysis. BrdU experiments were performed in triplicate wells for a total of 3 independent experiments. Western blot- 3 independent experiments were performed. Data are represented as mean with SEM, * $p < 0.05$, paired t test.

Page 20/21- "To confirm our mouse studies, *ACVR1* mutant and wildtype human cell lines were treated with LDN212854. *ACVR1* mutant human cell lines SU DIPG IV (*ACVR1* G328V; H3.1K27M) and DIPG 007 (*ACVR1* R206H; H3.3K27M) were more sensitive to LDN212854 treatment and had decreased cell proliferation at lower drug concentrations compared to *ACVR1* wildtype line SF8628 (Fig. 6G). Lysates from all three LDN212854 treated human lines also showed significantly decreased phosphorylated SMAD1/5/8 and Id1 protein levels (Fig. 6H and Supplemental Fig. 4F and G), indicating inhibition of the BMP pathway.

Furthermore, cell-lines from our *ACVR1* R206H; H3.1K27M; PDGF-A; p53 loss model were relatively resistant to treatment with LDN214117, another *ACVR1* inhibitor despite its ability to inhibit the BMP pathway (ID1 protein levels as the readout). Similarly, the human models were also resistant to LDN214117 even though the drug inhibited the BMP pathway with pSMAD 1/5/8 and ID1 protein levels as the readout. The results with LDN214117 in both the murine and human models are provided below and in Supplemental Figure 4H-N. We conclude that our mouse model may be a relevant preclinical model to help evaluate BMP pathway inhibitors. The revised text is pasted below:

Figure Legend- Proliferation of LDN214117 treated human lines (n = 3 independent experiments, triplicate wells). Data represented as mean with SEM. Western blot analysis and quantification of pSMAD1/5/8 and Id1 expression of LDN214117 treated human lines (n = 3 independent experiments). Data represented as mean with SEM. * p < 0.05, paired t test.

Page 21- "Mouse neurospheres and human cell lines were also treated with a second *ACVR1* inhibitor LDN214117, an inhibitor that is also a potent inhibitor of *ALK2*⁷. In mouse neurosphere lines, while treatment with LDN214117 did appear to inhibit the BMP pathway as evident by decreased Id1 expression, cell proliferation and viability were not significantly affected (Supplemental Fig. 4H-J). Similar results were also observed in LDN214117 treated human cell lines (Supplemental Fig. 4K-N).

The described phenotypic markers are not clearly consistent with DIPGs, making it difficult to generalize the results to human tumors with these mutations.

Author response- Thank you for your comment. As there have been very few publications with immunohistochemistry staining of human DIPGs, we fully appreciate the reviewer's concern. We have listed below evidence indicating whether the phenotypic markers used in our mouse modeling work are expressed in the human disease.

1. PhosphoH3-Serine 10 (a marker for proliferating cells in M phase)- This marker has been used to assess proliferating cells in human DIPG cell-lines by western blot⁸
2. Nestin- Nestin immunostaining has been previously performed by Ballester LY et al.⁹ and was noted to be positive in 25% of human DIPGs (6/24)
3. Olig2- Olig2 immunostaining has been previously reported by Ballester LY et al.⁹ and was noted to be positive in 92% of human DIPGs (22/24)
4. GFAP-GFAP immunostaining has been previously reported by Ballester LY et al.⁹ and was noted to be positive in 100% of human DIPGs (24/24)
5. CD44- CD44 has been noted to be expressed at the mRNA level of human DIPGs in at least two studies^{10,11} and in the latter, it has been noted to be expressed in DIPG associated microglia.
6. CD31- CD31 has been noted to be expressed by immunohistochemistry in a human DIPG xenograft model that harbors *ACVR1* G328V (DIPG IV)¹²
7. pStat3- In the revised manuscript and in this rebuttal letter, we have added western blots demonstrating increased Stat3 phosphorylation in three primary human DIPGs relative to matched normal brain.
8. Iba1- Iba1 has been noted to be expressed in a subset of human DIPG by immunofluorescence¹¹.
9. Vimentin- vimentin protein has been noted to be expressed by immunofluorescence in a DIPG cell-line⁸ and by immunohistochemistry in primary DIPGs tumor as well as xenograft models¹³
10. ID1- In the revised manuscript and in this rebuttal letter, we have added western blots of 3 human DIPG cell-lines treated with LDN212854 and LDN211417. ID1, a target gene of the BMP pathway is expressed in all 3 of the human DIPG lines and ID1 levels are reduced by treatment with the two *ACVR1* inhibitors.
11. Hes1- A subset of primary human DIPGs and cell-lines have been noted to express Hes1 protein (western blot) or mRNA¹⁴

We have added the following sentences in the text:

Page 12- "In an effort to further characterize the cells within the glioma-like lesions, we performed additional immunohistochemistry staining for Nestin, Olig2, and GFAP as all of these markers have been noted to be expressed in at least a subset of human DIPGs⁹ (Fig. 3a)."

Page 13- "Although immunostaining of human DIPGs for CD44 has not been reported, CD44 mRNA has been reported to be expressed in a subset of human DIPGs by two independent groups, one of which noted that CD44 is expressed in DIPG associated microglia^{10,11}"

Page 13- "Of note, both vimentin and Iba1 have been previously reported to stain human DIPGs^{11,13}."

Page 17- "Of note, Hes1 is also a component of the Notch pathway and while IHC for Hes1 has yet to be performed in human DIPGs, Hes1 has been reported to be expressed at both the mRNA and protein levels of human DIPGs¹⁴."

Reviewer #2 (Remarks to the Author):

In this manuscript by Hoeman et al, the authors describe experiments using mouse models to unravel the importance of mutations in ACVR1 and H3.1 in a subset of diffuse intrinsic pontine gliomas DIPG. DIPG is a highly lethal CNS tumor of children and in desperate need of better understanding. One of the recent observations is that histone mutations are found in the majority of these tumors. And one of these recurrent mutations shows co-occurrence of mutations in the gene encoding ALK2 (ACVR1). This paper uses in vivo mouse models and cell culture experiments to unravel this connection.

The experiments are well done, and highly detailed. The topic has general interest. There are a few things that would strengthen the paper:

Author's response- Thanks for the positive feedback.

Is there documentation that LDN212854 gets into the brain?

Author's response- Yes, we measured drug concentrations in the tumor 4 hours after the tenth dose with the goal of measuring drug at steady state (5 days of treatment with twice a day dosing). Such graph is depicted below and is in Figure 6K in the revised manuscript. We observed significant inhibition of the BMP pathway as well as significant reduction in proliferation *in vivo* at this time point suggesting that this concentration of drug is sufficient for antitumor effect even though it is a lower concentration than the IC50 observed in the *in vitro* studies (see Figure 6I or 6J)

Figure legend- LDN212854 drug levels measured in the cortex and tumor of drug (LDN212854) treated mice. Time-point was 4 hours post 10th dose (BID dosing X 5 days). Data are represented as mean with SEM.

Does the ACVR1 mutation do anything collaboratively with the other histone mutations? In cell culture? Looking for some evidence that the co-occurrence of this specific histone mutation makes sense.

Author's response- Thanks for your comment. We performed additional experiments to determine if *ACVR1* R206H *ACVR1* collaborates with the other histone mutation (H3.3K27M) and this new data has been added to Figure 4A and 4C and is also included in the next page (blue curve). Interestingly, we observed similar results when we used H3.3K27M in lieu of H3.1K27M. The *ACVR1* R206H; H3.3K27M; PDGF-A; p53 loss experimental group had a high tumor incidence (20/22 = 91%) similar to the tumor incidence with *ACVR1* R206H; H3.1K27M; PDGF-A; p53 loss (28/31 = 90%).

With regards to the question looking for evidence that the co-occurrence of H3.1K27M and *ACVR1* R206H makes sense- the most compelling evidence is the cooperation in tumor initiation as *ACVR1* R206H; H3.1K27M; PDGF-A; p53 loss have a significant higher tumor incidence (28/31=90%) than the following three cohorts: *ACVR1* WT; H3.1K27M; PDGF-A; p53 loss (10/17=59%; p=0.022(*)), *ACVR1* R206H; H3.1 WT; PDGF-A; p53 loss (11/25=44%; p=0.0003(***)), and *ACVR1* WT; H3.1 WT; PDGF-A; p53 loss (15/26=58%; p=0.0059(**)).

The revised text is pasted below the figures.

It is worth noting that in human patient samples, *ACVR1* R206H can partner with both H3.1K27M and H3.3K27M almost equally while the other *ACVR1* mutations in the kinase domain (e.g. G328 mutations) primarily partner with H3.1K27M. This suggests that *ACVR1* R206H and/or other GS domain mutations may behave differently than *ACVR1* mutations in the kinase domain. As most of the *in vivo* experiments in this manuscript are with the *ACVR1* R206H mutation (as it was most potent *in vitro*), we removed the *in vivo* data with the G328V as it may behave differently than the R206H mutation and perhaps will synergize best with H3.1K27M vs. H3.3K27M (as in the human disease). This will be the subject of future studies.

(A) Kaplan-Meier survival curve of Nestin *tv-a*; *p53^{fl/fl}* mice that were injected with RCAS-PDGFA, RCAS-Cre, RCAS-H3.1K27M, and RCAS-ACVR1 R206H (n = 31) or RCAS-ACVR1 WT (n = 17). For control purposes, mice were injected with RCAS-PDGFA, RCAS-Cre, RCAS-H3.3K27M, and RCAS-ACVR1 R206H (n = 23), RCAS-PDGFA, RCAS-Cre RCAS-H3.1 WT, and RCAS-ACVR1 R206H (n = 26), or RCAS-PDGFA, RCAS-Cre RCAS-H3.1 WT, and RCAS-ACVR1 WT (n = 26). For significant differences among groups see Table 1. * $p < 0.05$, log-rank test.

Tumor presence was assessed by H&E staining and confirmed by a blinded neuropathologist. For tumor incidence rates and significant differences among groups see Table 2. * $p < 0.05$, Fischer's exact test.

Page 15/16- "While examining the effect of *ACVR1* mutations on survival, we noted that mice that were not infected with *ACVR1* R206H were more likely to survive to the end of the study and remain asymptomatic (6 months post tumor initiation) as compared to mice that were infected with *ACVR1* mutation R206H. Therefore, we hypothesized that *ACVR1* R206H might have an effect on tumor incidence as well. In fact, mice that were infected with *ACVR1* R206H, H3.1K27M, Cre, and PDGFA had increased tumor incidence (28/31 = 90%) as compared to control mice that were infected with *ACVR1*

WT, H3.1K27M, Cre, and PDGFA (10/17 = 59%, * p = .0220) or ACVR1 WT, H3.1 WT, Cre, and PDGFA (15/26 = 58%, ** p = .0059) (Fig. 4c and Table 2). Surprisingly, mice that were injected with ACVR1 R206H, Cre, and PDGFA but not H3.1 K27M or ACVR1 R206H, H3.1 WT, Cre, and PDGFA did not demonstrate an increase in tumor incidence (21/26 = 81% and 11/25 = 44% respectively) compared to any group except each other (Fig. 4d and Table 2), suggesting that H3.1K27M is required for the effect of ACVR1 R206H on tumor initiation and is in agreement with our observations with the glioma-like lesions without PDGFA.”

On page 10 there are many statements of increasing of lesion size and incidence of formation. But with small numbers. And I suspect not statistically significant. Best to not overstate on limited data without statistics.

Author’s response- Thanks for your comment. We have removed all statements regarding increasing of lesion size and incidence on page 10. We completely agree with the reviewer.

In the analysis of the tumors and “glioma-like lesions”. Was co immunofluorescence done to verify that the actual tumor cells per se had the characteristics listed? Rather than stroma.

Author’s response- This is a great question. Unfortunately, the glioma-like lesions were very small yielding only a few unstained sections for immunostaining once we identified them by H&E. We confirmed the expression of the mutant proteins in these glioma-like lesions using immunofluorescence for the tags (note the nuclear HA staining for the mutant histone and cytoplasmic FLAG staining for the mutant ACVR1- See figure 2). The immunohistochemistry for phospho-histone3-serine 10, pStat3, Olig2 were done with adjacent sections so we cannot conclude definitely as to whether the cells that expressed H3.1K27M or R206H ACVR1 also expressed phospho-histone3-serine 10, or pStat3, or Olig2.

We added the following on the bottom of page 13- “It is worth noting that due to the small size of the lesions we were unable to perform double immunofluorescence to determine if the cells expressing H3.1K27M and mutant ACVR1 are also expressing pSTAT3.”

Could a more concrete delineation of actual DIPG from “glioma-like lesions” be stated. It seems that PDGF is needed for the actual cancer, whatever that means.

Author’s response- Thanks for your question. It is important to note that the glioma-like lesions were reviewed by an experienced neuropathologist (Roger McLendon) who was confident that these lesions were not DIPGs. We do not think that we can speculate whether the glioma-like lesions induced by ACVR1 R206H, H3.1K27M, and p53 loss

are early DIPG lesions (i.e. that they would have progressed to full DIPGs if we would have extended our observation time post infection with the above listed viruses). A limitation of our study is that we observed the mice for only six months' post infection with the above listed viruses. We can only state that we are the first to demonstrate that these three genetic alterations are sufficient to induce proliferation as well as expression of mesenchymal markers in the neonatal brainstem.

The requirement for PDGF suggests that PDGF signaling is required for DIPG pathogenesis with this modeling system, a postnatal modeling system. In the initial publications in 2014 reporting the presence of *ACVR1* mutations in approximately 25% of human DIPGs, two publications observed the presence PDGFRA amplifications in *ACVR1* mutant DIPGs: Wu et al. noted its presence in one *ACVR1* mutant/H3.1K27M mutant tumor and Buczkowicz et al. noted its presence in four *ACVR1* mutant tumors (two H3.1K27M mutant and two H3.3 K27M mutant)^{15,16}. Thus, PDGFRA signaling can be seen in human tumors with *ACVR1* mutants/ H3.1K27M although it is not common. More recently, PDGF-A has been reported to be expressed in at least a subset of DIPG tumor cells in at least 3 out of 4 different assays (cultured DIPG RNAseq, primary bulk DIPG RNAseq, primary single cell DIPG RNAseq)¹¹. Lastly, it is also possible that PDGFRA signaling in the tumor may also be activated in a paracrine fashion by stromal cells such as astrocytes¹⁷, endothelial cells¹⁸ or neurons¹⁹, all of which have been reported to secrete PDGF ligands as part of normal brain development and so this may be an important topic for future studies to investigate as several culture protocols for human DIPG cells include one of the PDGF ligands²⁰ which suggest that it has an important role in the human disease. We added the following in the discussion section on page 25:

“As mutant *ACVR1*, H3.1K27M, and p53 loss were not sufficient to induce murine DIPGs, we decided to add PDGF-A to the model as PDGFRA amplifications are occasionally seen in human DIPGs that harbor *ACVR1* mutations^{15,16}. While the significance of the requirement for PDGF-A to develop *ACVR1* mutant DIPGs with this modeling approach is unclear, PDGF-A has recently been reported to be expressed in at least a subset of DIPG tumor cells in at least 3 out of 4 different assays (cultured DIPG RNAseq, primary bulk DIPG RNAseq, primary single cell DIPG RNAseq)¹¹. It is possible that tumor initiation *in utero* with these genetic alterations or in a different cell-of-origin postnatally may obviate the requirement for PDGF-A ligand (or other similar genetic alteration such as mutant PDGFRA).”

Was the state of H3.1 in the tumors that didn't get the RCAS version of H3.1 analyzed? Is there a selection for the parallel alterations in the mouse version of this protein? Or was the AKT pathway activity analyzed in the tumors that didn't get the mutation in PI3K? again, maybe elevated activity was selected for when the tumor finally did arise.

Author's response- Thanks for your question. We have done RNAseq of tumors without the RCAS-H3.1K27M or with the RCAS-H3.1 WT and did not observe a spontaneous

K27M mutation in the endogenous H3.1 genes. We did not look at the AKT pathway in detail in this manuscript (except evaluating whether PTEN loss can substitute for the requirement for PDGF signaling and it cannot in this system). This will be the subject of future studies and so we removed the mutant PIK3CA data from this manuscript.

The paragraph on page 14 is a bit confusing, as is the one on page 15. Maybe a summary of the results could replace these paragraphs and the detailed text here could be moved to supplementary data.

Author's response- Thanks for your suggestion. We have clarified the text and reference several tables in the text that summarize the results for readers that prefer a table format to review our observations. Please see table 1-4 and supplemental table 3.

On page 18 there seems to be 5 vectors all mixed together and infected mice. And one of them is luciferase. How sure are the authors that all 5 vectors got into the tumor cells, especially since luciferase is immunogenic when transferred. Germline luciferase when activated with Cre is less so I understand.

Author's response- Thanks for your suggestion. We will definitely consider using germline luciferase in future studies. The experiments with the 5 vectors were done for only one part of the study (short-term treatment with LDN212854 where we wanted to identify tumor-bearing mice before they developed symptoms for the pharmacological treatment studies). For the survival study, we ended up switching back to 4 vectors and we used MRI instead to image tumors as the luciferase signal was not 100% sensitive and specific to identify tumor-bearing mice. We have done RNAseq on a small subset of the tumors that express 4 vectors (without luciferase) and know that all 4 vectors are expressed in most of the tumors (the *in vivo* RNAseq data will be part of a follow-up manuscript). Two vectors are absolutely required for the generation of large tumors that can be visualized by eye when extracting the brain from the mouse after euthanasia and to elicit brain tumor related symptoms in the mouse and those are PDGF-A and p53 deletion.

References

- 1 Pathania, M. *et al.* H3.3(K27M) Cooperates with Trp53 Loss and PDGFRA Gain in Mouse Embryonic Neural Progenitor Cells to Induce Invasive High-Grade Gliomas. *Cancer Cell* **32**, 684-700 e689, doi:10.1016/j.ccell.2017.09.014 (2017).
- 2 Funato, K., Major, T., Lewis, P. W., Allis, C. D. & Tabar, V. Use of human embryonic stem cells to model pediatric gliomas with H3.3K27M histone mutation. *Science* **346**, 1529-1533, doi:10.1126/science.1253799 (2014).

- 3 Cordero, F. J. *et al.* Histone H3.K27M Represses p16 to Accelerate Gliomagenesis in a Murine Model of DIPG. *Mol Cancer Res* **15**, 1243-1254, doi:10.1158/1541-7786.MCR-16-0389 (2017).
- 4 Carro, M. S. *et al.* The transcriptional network for mesenchymal transformation of brain tumours. *Nature* **463**, 318-325, doi:10.1038/nature08712 (2010).
- 5 Castel, D. *et al.* Histone H3F3A and HIST1H3B K27M mutations define two subgroups of diffuse intrinsic pontine gliomas with different prognosis and phenotypes. *Acta Neuropathol* **130**, 815-827, doi:10.1007/s00401-015-1478-0 (2015).
- 6 Castel, D. *et al.* Transcriptomic and epigenetic profiling of 'diffuse midline gliomas, H3 K27M-mutant' discriminate two subgroups based on the type of histone H3 mutated and not supratentorial or infratentorial location. *Acta Neuropathol Commun* **6**, 117, doi:10.1186/s40478-018-0614-1 (2018).
- 7 Mohedas, A. H. *et al.* Structure-activity relationship of 3,5-diaryl-2-aminopyridine ALK2 inhibitors reveals unaltered binding affinity for fibrodysplasia ossificans progressiva causing mutants. *J Med Chem* **57**, 7900-7915, doi:10.1021/jm501177w (2014).
- 8 Kumar, S. S. *et al.* BMI-1 is a potential therapeutic target in diffuse intrinsic pontine glioma. *Oncotarget* **8**, 62962-62975, doi:10.18632/oncotarget.18002 (2017).
- 9 Ballester, L. Y. *et al.* Morphologic characteristics and immunohistochemical profile of diffuse intrinsic pontine gliomas. *Am J Surg Pathol* **37**, 1357-1364, doi:10.1097/PAS.0b013e318294e817 (2013).
- 10 Puget, S. *et al.* Mesenchymal transition and PDGFRA amplification/mutation are key distinct oncogenic events in pediatric diffuse intrinsic pontine gliomas. *PLoS One* **7**, e30313, doi:10.1371/journal.pone.0030313 (2012).
- 11 Lin, G. L. *et al.* Non-inflammatory tumor microenvironment of diffuse intrinsic pontine glioma. *Acta Neuropathol Commun* **6**, 51, doi:10.1186/s40478-018-0553-x (2018).
- 12 Shaik, S. *et al.* REST upregulates gremlin to modulate diffuse intrinsic pontine glioma vasculature. *Oncotarget* **9**, 5233-5250, doi:10.18632/oncotarget.23750 (2018).
- 13 Plessier, A. *et al.* New in vivo avatars of diffuse intrinsic pontine gliomas (DIPG) from stereotactic biopsies performed at diagnosis. *Oncotarget* **8**, 52543-52559, doi:10.18632/oncotarget.15002 (2017).
- 14 Taylor, I. C. *et al.* Disrupting NOTCH Slows Diffuse Intrinsic Pontine Glioma Growth, Enhances Radiation Sensitivity, and Shows Combinatorial Efficacy With Bromodomain Inhibition. *J Neuropathol Exp Neurol* **74**, 778-790, doi:10.1097/NEN.0000000000000216 (2015).
- 15 Wu, G. *et al.* The genomic landscape of diffuse intrinsic pontine glioma and pediatric non-brainstem high-grade glioma. *Nat Genet* **46**, 444-450, doi:10.1038/ng.2938 (2014).
- 16 Buczkowicz, P. *et al.* Genomic analysis of diffuse intrinsic pontine gliomas identifies three molecular subgroups and recurrent activating ACVR1 mutations. *Nat Genet* **46**, 451-456, doi:10.1038/ng.2936 (2014).
- 17 Richardson, W. D., Pringle, N., Mosley, M. J., Westermarck, B. & Dubois-Dalcq, M. A role for platelet-derived growth factor in normal gliogenesis in the central nervous system. *Cell* **53**, 309-319 (1988).
- 18 Andrae, J., Gallini, R. & Betsholtz, C. Role of platelet-derived growth factors in physiology and medicine. *Genes Dev* **22**, 1276-1312, doi:10.1101/gad.1653708 (2008).

- 19 Fruttiger, M., Calver, A. R. & Richardson, W. D. Platelet-derived growth factor is constitutively secreted from neuronal cell bodies but not from axons. *Curr Biol* **10**, 1283-1286 (2000).
- 20 Monje, M. *et al.* Hedgehog-responsive candidate cell of origin for diffuse intrinsic pontine glioma. *Proc Natl Acad Sci U S A* **108**, 4453-4458, doi:10.1073/pnas.1101657108 (2011).

REVIEWERS' COMMENTS:

Reviewer #1 (Remarks to the Author):

In this revised manuscript, the authors provide further evidence to strengthen the argument that Stat3 signaling and EMT expression are enhanced by ACVR1 mutations in the context of H3.1K27M in nestin expressing brainstem progenitors, and show evidence that LDN212854 is inhibitory in human DIPG cell lines. The work is done well and will be of interest to the general research community. My concerns have been adequately addressed.

Reviewer #2 (Remarks to the Author):

The authors have answered my questions adequately.